# Optimal Order Simple Regret for Gaussian Process Bandits

**Sattar Vakili**[*], **Nacime Bouziani**[+], **Sepehr Jalali**[*], **Alberto Bernacchia**[*], **Da-shan Shiu**[*]

[*] MediaTek Research

{sattar.vakili, sepehr.jalali, alberto.bernacchia, ds.shiu}@mtkresearch.com

[+]Imperial College London

n.bouziani18@imperial.ac.uk

## Abstract

Consider the sequential optimization of a continuous, possibly non-convex, and expensive to evaluate objective function $f$. The problem can be cast as a Gaussian Process (GP) bandit where $f$ lives in a reproducing kernel Hilbert space (RKHS). The state of the art analysis of several learning algorithms shows a significant gap between the lower and upper bounds on the simple regret performance. When $N$ is the number of exploration trials and $\gamma_N$ is the maximal information gain, we prove an $\tilde{\mathcal{O}}(\sqrt{\gamma_N/N})$ bound on the simple regret performance of a pure exploration algorithm that is significantly tighter than the existing bounds. We show that this bound is order optimal up to logarithmic factors for the cases where a lower bound on regret is known. To establish these results, we prove novel and sharp confidence intervals for GP models applicable to RKHS elements which may be of broader interest.

## 1 Introduction

Sequential optimization has evolved into one of the fastest developing areas of machine learning [1]. We consider sequential optimization of an unknown objective function from noisy and expensive to evaluate zeroth-order[1] observations. That is a ubiquitous problem in academic research and industrial production. Examples of applications include exploration in reinforcement learning, recommendation systems, medical analysis tools and speech recognizers [4]. A notable application in the field of machine learning is automated hyper-parameter tuning. Prevalent methods such as grid search can be prohibitively expensive [5, 6]. Sequential optimization methods, on the other hand, are shown to efficiently find good hyper-parameters by an adaptive exploration of the hyper-parameter space [7].

Our sequential optimization setting is as follows. Consider an objective function $f$ defined over a domain $\mathcal{X} \subset \mathbb{R}^d$, where $d \in \mathbb{N}$ is the dimension of the input. A learning algorithm is allowed to perform an adaptive exploration to sequentially observe the potentially corrupted values of the objective function $\{f(x_n) + \epsilon_n\}_{n=1}^N$, where $\epsilon_n$ are random noises. At the end of $N$ exploration trials, the learning algorithm returns a candidate maximizer $\hat{x}_N^* \in \mathcal{X}$ of $f$. Let $x^* \in \operatorname{argmax}_{x \in \mathcal{X}} f(x)$ be a true optimal solution. We may measure the performance of the learning algorithm in terms of *simple regret*; that is, the difference between the performance under the true optimal solution, $f(x^*)$, and that under the learnt value, $f(\hat{x}_N^*)$.

Our formulation falls under the general framework of continuum armed bandit that signifies receiving feedback only for the selected observation point $x_n$ at each time $n$ [8, 9, 10, 11]. Bandit problems have been extensively studied under numerous settings and various performance measures including simple

---

[1]Zeroth-order feedback signifies observations from $f$ in contrast to first-order feedback which refers to observations from gradient of $f$ as e.g. in stochastic gradient descent [see, e.g., 2, 3].

35th Conference on Neural Information Processing Systems (NeurIPS 2021).

regret [see, e.g., 10, 12, 13], cumulative regret [see, e.g., 14, 15, 16], and best arm identification [see, e.g., 17, 18]. The choice of performance measure strongly depends on the application. Simple regret is suitable for situations with a preliminary exploration phase (for instance hyper-parameter tuning) in which costs are not measured in terms of rewards but rather in terms of resources expended [10].

Due to infinite cardinality of the domain, approaching $f(x^*)$ is feasible only when appropriate regularity assumptions on $f$ and noise are satisfied. Following a growing literature [19, 20, 21, 22], we focus on a variation of the problem where $f$ is assumed to belong to a reproducing kernel Hilbert space (RKHS) that is a very general assumption. Almost all continuous functions can be approximated with the RKHS elements of practically relevant kernels such as Matérn family of kernels [19]. We consider two classes of noise: sub-Gaussian and light-tailed.

Our regularity assumption on $f$ allows us to utilize Gaussian processes (GPs) which provide powerful Bayesian (surrogate) models for $f$ [23]. Sequential optimization based on GP models is often referred to as Bayesian optimization in the literature [4, 24, 25]. We build on prediction and uncertainty estimates provided by GP models to study an efficient adaptive exploration algorithm referred to as Maximum Variance Reduction (MVR). Under simple regret measure, MVR embodies the simple principle of exploring the points with the highest variance first. Intuitively, the variance in the GP model is considered as a measure of uncertainty about the unknown objective function and the exploration steps are designed to maximally reduce the uncertainty. At the end of exploration trials, MVR returns a candidate maximizer based on the prediction provided by the learnt GP model. With its simple structure, MVR is amenable to a tight analysis that significantly improves the best known bounds on simple regret. To this end, we derive novel and sharp confidence intervals for GP models applicable to RKHS elements. In addition, we provide numerical experiments on the simple regret performance of MVR comparing it to GP-UCB [19, 20], GP-PI [26] and GP-EI [26].

## 1.1 Main Results

We first derive novel confidence intervals for GP models applicable to RKHS elements (Theorems 1 and 2). As part of our analysis, we formulate the posterior variance of a GP model as the sum of two terms: the maximum prediction error from noise-free observations, and the effect of noise (Proposition 1). This interpretation elicits new connections between GP regression and kernel ridge regression [27]. These results are of interest on their own.

We then build on the confidence intervals for GP models to provide a tight analysis of the simple regret of the MVR algorithm (Theorem 3). In particular, we prove a high probability $\tilde{\mathcal{O}}(\sqrt{\frac{\gamma_N}{N}})^2$ simple regret, where $\gamma_N$ is the maximal information gain (see § 2.4). In comparison to $\tilde{\mathcal{O}}(\frac{\gamma_N}{\sqrt{N}})$ bounds on simple regret [see, e.g., 19, 20, 28], we show an $\mathcal{O}(\sqrt{\gamma_N})$ improvement. It is noteworthy that our bound guarantees convergence to the optimum value of $f$, while previous $\tilde{\mathcal{O}}(\frac{\gamma_N}{\sqrt{N}})$ bounds do not, since although $\gamma_N$ grows sublinearly with $N$, it can grow faster than $\sqrt{N}$.

We then specialize our results for the particular cases of practically relevant Matérn and Squared Exponential (SE) kernels. We show that our regret bounds match the lower bounds and close the gap reported in [28, 29], who showed that an average simple regret of $\epsilon$ requires $N = \Omega\left(\frac{1}{\epsilon^2}(\log(\frac{1}{\epsilon}))^{\frac{d}{2}}\right)$ exploration trials in the case of SE kernel. For the Matérn-$\nu$ kernel (where $\nu$ is the smoothness parameter, see § 2.1) they gave the analogous bound of $N = \Omega\left((\frac{1}{\epsilon})^{2+\frac{d}{\nu}}\right)$. They also reported a significant gap between these lower bounds and the upper bounds achieved by GP-UCB algorithm. In Corollary 1, we show that our analysis of MVR closes this gap in the performance and establishes upper bounds matching the lower bounds up to logarithmic factors.

In contrast to the existing results which mainly focus on Gaussian and sub-Gaussian distributions for noise, we extend our analysis to the more general class of light-tailed distributions, thus broadening the applicability of the results. This extension increases both the confidence interval width and the simple regret by only a multiplicative logarithmic factor. These results apply to e.g. the privacy preserving setting where often a light-tailed noise is employed [30, 31, 32].

---

[2]The notations $\mathcal{O}$ and $\tilde{\mathcal{O}}$ are used to denote the mathematical order and the mathematical order up to logarithmic factors, respectively.

## 1.2 Literature Review

The celebrated work of Srinivas *et al.* [19] pioneered the analysis of GP bandits by proving an $\tilde{\mathcal{O}}(\gamma_N\sqrt{N})$ upper bound on the cumulative regret of GP-UCB, an optimistic optimization algorithm sequentially selecting $x_n$ which maximize an upper confidence bound index over the search space. That implies an $\tilde{\mathcal{O}}(\frac{\gamma_N}{\sqrt{N}})$ simple regret [28]. Their analysis relied on deriving confidence intervals for GP models applicable to RKHS elements. They also considered a fully Bayesian setting where $f$ is assumed to be a sample from a GP and noise is assumed to be Gaussian. [20] built on feature space representation of GP models and self-normalized martingale inequalities, first developed in [33] for linear bandits, to improve the confidence intervals of [19] by a multiplicative $\log(N)$ factor. That led to an improvement in the regret bounds by the same multiplicative $\log(N)$ factor. A discussion on the comparison between these results and the confidence intervals derived in this paper is provided in § 3.3. A technical comparison with some recent advances in regret bounds requires introducing new notations and is deferred to § 4.4.

The performance of Bayesian optimization algorithms has been extensively studied under numerous settings including contextual information [34], high dimensional spaces [35, 36], safety constraints [37, 38], parallelization [39], meta-learning [40], multi-fidelity evaluations [41], ordinal models [42], corruption tolerance [43, 29], and neural tangent kernels [44, 45]. [46] introduced an adaptive discretization of the search space improving the computational complexity of a GP-UCB based algorithm. Sparse approximation of GP posteriors are shown to preserve the regret orders while improving the computational complexity of Bayesian optimization algorithms [36, 47, 48]. Under the RKHS setting with noisy observations, GP-TS [20] and GP-EI [49, 50] are also shown to achieve the same regret guarantees as GP-UCB (up to logarithmic factors). All these works report $\tilde{\mathcal{O}}(\frac{\gamma_N}{\sqrt{N}})$ regret bounds.

The regret bounds are also reported under other often simpler settings such as noise-free observations [51, 52, $\epsilon_n = 0, \forall n$] or a Bayesian regret that is averaged over a known prior on $f$ [39, 53, 54, 55, 56, 57, 58, 59], rather than for a fixed and unknown $f$ as in our setting.

Other lines of work on continuum armed bandit exist, relying on other regularity assumptions such as Lipschitz continuity [9, 11, 12, 60], convexity [61] and unimodality [62], to name a few. A notable example is [11] who showed that hierarchical algorithms based on tree search yield $\mathcal{O}(N^{\frac{d+1}{d+2}})$ cumulative regret. We do not compare with these results due to the inherent difference in the regularity assumptions.

## 1.3 Organization

In § 2, the problem formulation, the regularity assumptions, and the preliminaries on RKHS and GP models are presented. The novel confidence intervals for GP models are proven in § 3. MVR algorithm and its analysis are given in § 4. The experiments are presented in § 5. We conclude with a discussion in § 6.

## 2 Problem Formulation and Preliminaries

Consider an objective function $f : \mathcal{X} \rightarrow \mathbb{R}$, where $\mathcal{X} \subseteq \mathbb{R}^d$ is a convex and compact domain. Consider an optimal point $x^* \in \text{argmax}_{x \in \mathcal{X}} f(x)$. A learning algorithm $\mathcal{A}$ sequentially selects observation points $\{x_n \in \mathcal{X}\}_{n \in \mathbb{N}}$ and observes the corresponding noise disturbed objective values $\{y_n = f(x_n) + \epsilon_n\}_{n \in \mathbb{N}}$, where $\epsilon_n$ is the observation noise. We use the notations $\mathcal{H}_n = \{X_n, Y_n\}$, $X_n = [x_1, x_2, ..., x_n]^\top$, $Y_n = [y_1, y_2, ..., y_n]^\top$, $x_n \in \mathcal{X}$, $y_n \in \mathbb{R}$, for all $n \geq 1$. In a simple regret setting, the learning algorithm determines a sequence of mappings $\{\mathcal{S}_n\}_{n \geq 1}$ where each mapping $\mathcal{S}_n : \mathcal{H}_n \rightarrow \mathcal{X}$ predicts a candidate maximizer $\hat{x}_n^*$. For algorithm $\mathcal{A}$, the simple regret under a budget of $N$ tries is defined as

$$r_N^\mathcal{A} = f(x^*) - f(\hat{x}_N^*). \tag{1}$$

The budget $N$ may be unknown *a priori*. Notationwise, we use $F_n = [f(x_1), f(x_2), \ldots, f(x_n)]^\top$ and $E_n = [\epsilon_1, \epsilon_2, \ldots, \epsilon_n]^\top$ to denote the noise free part of the observations and the noise history, respectively, similar to $X_n$ and $Y_n$.

## 2.1 Gaussian Processes

The Bayesian optimization algorithms build on GP (surrogate) models. A GP is a random process $\{\hat{f}(x)\}_{x \in \mathcal{X}}$, where each of its finite subsets follow a multivariate Gaussian distribution. The distribution of a GP is fully specified by its mean function $\mu(x) = \mathbb{E}[\hat{f}(x)]$ and a positive definite kernel (or covariance function) $k(x, x') = \mathbb{E}\left[(\hat{f}(x) - \mu(x))(\hat{f}(x') - \mu(x'))\right]$. Without loss of generality, it is typically assumed that, for prior GP distributions, $\mu(x) = 0, \forall x \in \mathcal{X}$.

Conditioning GPs on available observations provides us with powerful non-parametric Bayesian (surrogate) models over the space of functions. In particular, using the conjugate property, conditioned on $\mathcal{H}_n$, the posterior of $\hat{f}$ is a GP with mean function $\mu_n(x) = \mathbb{E}[\hat{f}(x)|\mathcal{H}_n]$ and kernel function $k_n(x, x') = \mathbb{E}[(\hat{f}(x) - \mu_n(x))(\hat{f}(x') - \mu_n(x'))|\mathcal{H}_n]$ specified as follows:

$$
\begin{aligned}
\mu_n(x) &= k^\top(x, X_n)\left(k(X_n, X_n) + \lambda^2 I_n\right)^{-1} Y_n, \\
k_n(x, x') &= k(x, x') - k^\top(x, X_n)\left(k(X_n, X_n) + \lambda^2 I_n\right)^{-1} k(x', X_n), \ \sigma_n^2(x) \triangleq k_n(x, x), (2)
\end{aligned}
$$

where with some abuse of notation $k(x, X_n) = [k(x, x_1), k(x, x_2), \ldots, k(x, x_n)]^\top$, $k(X_n, X_n) = [k(x_i, x_j)]_{i,j=1}^n$ is the covariance matrix, $I_n$ is the identity matrix of dimension $n$, and $\lambda > 0$ is a real number.

In practice, Matérn and squared exponential (SE) are the most commonly used kernels for Bayesian optimization [see, e.g., 4, 24],

$$
k_{\text{Matérn}}(x, x') = \frac{1}{\Gamma(\nu)2^{\nu-1}}\left(\frac{\sqrt{2\nu}\rho}{l}\right)^\nu B_\nu\left(\frac{\sqrt{2\nu}\rho}{l}\right), \quad k_{\text{SE}}(x, x') = \exp\left(-\frac{\rho^2}{2l^2}\right),
$$

where $l > 0$ is referred to as lengthscale, $\rho = ||x - x'||_{l_2}$ is the Euclidean distance between $x$ and $x'$, $\nu > 0$ is referred to as the smoothness parameter, $\Gamma$ and $B_\nu$ are, respectively, the Gamma function and the modified Bessel function of the second kind. Variation over parameter $\nu$ creates a rich family of kernels. The SE kernel can also be interpreted as a special case of Matérn family when $\nu \to \infty$.

## 2.2 RKHSs and Regularity Assumptions on $f$

Consider a positive definite kernel $k : \mathcal{X} \times \mathcal{X} \to \mathbb{R}$ with respect to a finite Borel measure (e.g., the Lebesgue measure) supported on $\mathcal{X}$. A Hilbert space $H_k$ of functions on $\mathcal{X}$ equipped with an inner product $\langle \cdot, \cdot \rangle_{H_k}$ is called an RKHS with reproducing kernel $k$ if the following is satisfied. For all $x \in \mathcal{X}$, $k(\cdot, x) \in H_k$, and for all $x \in \mathcal{X}$ and $f \in H_k$, $\langle f, k(\cdot, x) \rangle_{H_k} = f(x)$ (reproducing property). A constructive definition of RKHS requires introducing Mercer theorem, which provides an alternative representation of kernels as an inner product of infinite dimensional feature maps [see, e.g., 27, Theorem 4.1], and is deferred to Appendix A. We have the following regularity assumption on the objective function $f$.

**Assumption 1** *The objective function $f$ is assumed to live in the RKHS corresponding to a positive definite kernel $k$. In particular, $||f||_{H_k} \leq B$, for some $B > 0$, where $||f||_{H_k}^2 = \langle f, f \rangle_{H_k}$.*

For common kernels, such as Matérn family of kernels, members of $H_k$ can uniformly approximate any continuous function on any compact subset of the domain $\mathcal{X}$ [19]. This is a very general class of functions; more general than, e.g., the class of convex functions. It has thus gained increasing interest in recent years.

## 2.3 Regularity Assumptions on Noise

We consider two different cases regarding the regularity assumption on noise. Let us first revisit the definition of sub-Gaussian distributions.

**Definition 1** *A random variable $X$ is called sub-Gaussian if its moment generating function $M(h) \triangleq \mathbb{E}[\exp(hX)]$ is upper bounded by that of a Gaussian random variable.*

The sub-Gaussian assumption implies that $\mathbb{E}[X] = 0$. It also allows us to use Chernoff-Hoeffding concentration inequality [63] in our analysis.

We next recall the definition of light-tailed distributions.

**Definition 2** *A random variable $X$ is called light-tailed if its moment-generating function exists, i.e., there exists $h_0 > 0$ such that for all $|h| \leq h_0$, $M(h) < \infty$.*

For a zero mean light-tailed random variable $X$, we have [64]

$$M(h) \quad \leq \quad \exp(\xi_0 h^2/2), \ \forall |h| \leq h_0, \xi_0 = \sup\{M^{(2)}(h), |h| \leq h_0\}, \tag{3}$$

where $M^{(2)}(.)$ denotes the second derivative of $M(.)$ and $h_0$ is the parameter specified in Definition 2. We observe that the upper bound in (3) is the moment generating function of a zero mean Gaussian random variable with variance $\xi_0$. Thus, light-tailed distributions are also called locally sub-Gaussian distributions [65].

We provide confidence intervals for GP models and regret bounds for MVR under each of the following assumptions on the noise terms.

**Assumption 2 (Sub-Gaussian Noise)** *The noise terms $\epsilon_n$ are independent over $n$. In addition, $\forall h \in \mathbb{R}, \forall n \in \mathbb{N}, \mathbb{E}[e^{h\epsilon_n}] \leq \exp(\frac{h^2 R^2}{2})$, for some $R > 0$.*

**Assumption 3 (Light-Tailed Noise)** *The noise terms $\epsilon_n$ are zero mean independent random variables over $n$. In addition, $\forall h \leq h_0, \forall n \in \mathbb{N}, \mathbb{E}[e^{h\epsilon_n}] \leq \exp(\frac{h^2 \xi_0}{2})$, for some $\xi_0 > 0$.*

Bayesian optimization uses GP priors for the objective function $f$ and assumes a Gaussian distribution for noise (for its conjugate property). It is noteworthy that the use of GP models is merely for the purpose of algorithm design and does not affect our regularity assumptions on $f$ and noise. We use the notation $\hat{f}$ to distinguish the GP model from the fixed $f$.

## 2.4 Maximal Information Gain

The regret bounds derived in this work are given in terms of the maximal information gain, defined as $\gamma_N = \sup_{X_N \subseteq \mathcal{X}} \mathcal{I}(Y_N; \hat{f})$, where $\mathcal{I}(Y_n; \hat{f})$ denotes the mutual information between $Y_n$ and $\hat{f}$ [see, e.g., 66, Chapter 2]. In the case of a GP model, the mutual information can be given as $\mathcal{I}(Y_n; \hat{f}) = \frac{1}{2} \log \det \left( I_n + \frac{1}{\lambda^2} k(X_n, X_n) \right)$, where the notation $(\log) \det$ denotes the (logarithm of) determinant of a square matrix. Note that the maximal information gain is kernel-specific and $X_N$-independent. Upper bounds on $\gamma_N$ are derived in [19, 21, 22] which are commonly used to provide explicit regret bounds. In the case of Matérn and SE kernels, $\gamma_N = \mathcal{O}\left( N^{\frac{d}{2\nu+d}} (\log(N))^{\frac{2\nu}{2\nu+d}} \right)$ and $\gamma_N = \mathcal{O}\left( (\log(N))^{d+1} \right)$, respectively [22].

## 3 Confidence Intervals for Gaussian Process Models

The analysis of bandit problems classically builds on confidence intervals applicable to the values of the objective function [see, e.g., 67, 68]. The GP modelling allows us to create confidence intervals for complex functions over continuous domains. In particular, we utilize the prediction ($\mu_n$) and the uncertainty estimate ($\sigma_n$) provided by GP models in building the confidence intervals which become an important building block of our analysis in the next section. To this end, we first prove the following proposition which formulates the posterior variance of a GP model as the sum of two terms: the maximum prediction error for an RKHS element from noise free observations and the effect of noise.

**Proposition 1** *Let $\sigma_n^2$ be the posterior variance of the surrogate GP model as defined in (2). Let $Z_n^\top(x) = k^\top(x, X_n) \left( k(X_n, X_n) + \lambda^2 I_n \right)^{-1}$. We have*

$$\sigma_n^2(x) = \sup_{f:||f||_{H_k} \leq 1} (f(x) - Z_n^\top(x) F_n)^2 + \lambda^2 \|Z_n(x)\|_{l^2}^2.$$

Notice that the first term $f(x) - Z_n^\top(x) F_n$ captures the maximum prediction error from noise free observations $F_n$. The second term captures the effect of noise in the surrogate GP model (and is independent of $F_n$). A detailed proof for Proposition 1 is provided in Appendix B.

Proposition 1 elicits new connections between GP models and kernel ridge regression. While the equivalence of the posterior mean in GP models and the regressor in kernel ridge regression is well known, the interpretation of posterior variance of GP models as the maximum prediction error for an RKHS element is less studied [see 27, Section 3, for a detailed discussion on the connections between GP models and kernel ridge regression].

## 3.1 Confidence Intervals under Sub-Gaussian Noise

The following theorem provides a confidence interval for GP models applicable to RKHS elements under the assumption that the noise terms are sub-Gaussian.

**Theorem 1** *Assume Assumptions 1 and 2 hold. Provided $n$ noisy observations $\mathcal{H}_n = \{X_n, Y_n\}$ from $f$, let $\mu_n$ and $\sigma_n$ be as defined in (2). Assume $X_n$ are independent of $E_n$. For a fixed $x \in \mathcal{X}$, define the upper and lower confidence bounds, respectively,*

$$U_n^\delta(x) \triangleq \mu_n(x) + (B + \beta(\delta))\,\sigma_n(x), \text{ and } L_n^\delta(x) \triangleq \mu_n(x) - (B + \beta(\delta))\,\sigma_n(x), \quad (4)$$

*with $\beta(\delta) = \frac{R}{\lambda}\sqrt{2\log(\frac{1}{\delta})}$, where $\delta \in (0,1)$, and $B$ and $R$ are the parameters specified in Assumptions 1 and 2. We have*

$$f(x) \leq U_n^\delta(x) \text{ w.p. at least } 1 - \delta, \text{ and } f(x) \geq L_n^\delta(x) \text{ w.p. at least } 1 - \delta.$$

We can write the difference in the objective function and the posterior mean as follows.

$$f(x) - \mu_n(x) = f(x) - Z_n^\top(x)Y_n = \underbrace{f(x) - Z_n^\top(x)F_n}_{\text{Prediction error from noise free observations}} - \underbrace{Z_n^\top(x)E_n}_{\text{The effect of noise}}.$$

The first term can be bounded directly following Proposition 1. The second term is bounded as a result of Proposition 1 and Chernoff-Hoeffding inequality. A detailed proof of Theorem 1 is provided in Appendix C.

## 3.2 Confidence Intervals under Light-Tailed Noise

We now extend the confidence intervals to the case of light-tailed noise. The main difference with sub-Gaussian noise is that Chernoff-Hoeffding inequality is no more applicable. We derive new bounds accounting for light-tailed noise in the analysis of Theorem 2.

**Theorem 2** *Assume Assumptions 1 and 3 hold. For a fixed $x \in \mathcal{X}$, define the upper and lower confidence bounds $U_n^\delta(x)$ and $L_n^\delta(x)$ similar to Theorem 1 with $\beta(\delta) = \frac{1}{\lambda}\sqrt{2\left(\xi_0 \vee \frac{2\log(\frac{1}{\delta})}{h_0^2}\right)\log(\frac{1}{\delta})}$ [3], where $\delta \in (0,1)$, and $B$, $h_0$ and $\xi_0$ are specified in Assumptions 1 and 3. Assume $X_n$ are independent of $E_n$. We have*

$$f(x) \leq U_n^\delta(x) \text{ w.p. at least } 1 - \delta, \text{ and } f(x) \geq L_n^\delta(x) \text{ w.p. at least } 1 - \delta.$$

In comparison to Theorem 1, under the light-tailed assumption, the confidence interval width increases with a multiplicative $\mathcal{O}(\sqrt{\log(\frac{1}{\delta})})$ factor. A detailed proof of Theorem 2 is provided in Appendix C.

**Remark 1** *Theorems 1 and 2 rely on the assumption that $X_n$ are independent of $E_n$. As we shall see in § 4, this assumption is satisfied when the confidence intervals are applied to the analysis of MVR.*

## 3.3 Comparison with the Existing Confidence Intervals

The most relevant work to our Theorems 1 and 2 is [20, Theorem 2] which itself was an improvement over [19, Theorem 6]. [20] built on feature space representation of GP kernels and self-normalized martingale inequalities [33, 69] to establish a $1 - \delta$ confidence interval in the same form as in Theorem 1, under Assumptions 1 and 2, with confidence interval width $B + R\sqrt{2(\gamma_n + 1 + \log(\frac{1}{\delta}))}$ [4]

---

[3]The notation $\vee$ is used to denote the maximum of two real numbers, $\forall a, b \in \mathbb{R}, (a \vee b) \triangleq \max(a, b)$.

[4]Note that the effect of $\lambda$ is absorbed in $\gamma_n$.

(instead of $B + \beta(\delta)$). There is a stark contrast between this confidence interval and the one given in Theorem 1 in its dependence on $\gamma_n$, which has a relatively large and possibly polynomial in $n$ value. That contributes an extra $\mathcal{O}(\sqrt{\gamma_N})$ multiplicative factor to regret.

Neither of these two results (our Theorem 1 and [20, Theorem 2]) imply the other. Although our confidence interval is much tighter, there are two important differences in the settings of these theorems. One difference is in the probabilistic dependencies between the observation points $x_n$ and the noise terms $\{\epsilon_j\}_{j<n}$. While Theorem 1 assumes that $X_n$ are independent of $E_n$, [20, Theorem 2] allows for the dependence of $x_n$ on the previous noise terms $\{\epsilon_j\}_{j<n}$. This is a reflection of the difference in the analytical requirements of MVR and GP-UCB. The other difference is that [20, Theorem 2] holds for all $x \in \mathcal{X}$. While, Theorem 1 holds for a single $x \in \mathcal{X}$. As we will see in § 4.2, a probability union bound can be used to obtain confidence intervals applicable to all $x$ in (a discretization of) $\mathcal{X}$, which contributes only logarithmic terms to regret, in contrast to $\mathcal{O}(\sqrt{\gamma_n})$. Roughly speaking, we are trading off the extra $\mathcal{O}(\sqrt{\gamma_n})$ term for restricting the confidence interval to hold for a single $x$. It remains an open problem whether the same can be done when $x_n$ are allowed to depend on $\{\epsilon_j\}_{j<n}$.

## 4 Maximum Variance Reduction and Simple Regret

In this section, we first formally present an exploration policy based on GP models referred to as Maximum Variance Reduction (MVR). We then utilize the confidence intervals for GP models derived in § 3 to prove bounds on the simple regret of MVR.

### 4.1 Maximum Variance Reduction Algorithm

MVR relies on the principle of maximally reducing the uncertainty, where the uncertainty is measured by the posterior variance of the GP model. After $N$ exploration trials, MVR returns a candidate maximizer according to the prediction provided by the learnt GP model. A pseudo-code is given in Algorithm 1.

---
**Algorithm 1** Maximum Variance Reduction (MVR)
---
1: **Initialization:** $k$, $\mathcal{X}$, $f$, $\sigma_0^2(x) \triangleq k(x,x)$.
2: **for** $n = 1, 2, \ldots, N$ **do**
3:      $x_n = \text{argmax}_{x \in \mathcal{X}} \sigma_{n-1}^2(x)$, where a tie is broken arbitrarily.
4:      Update $\sigma_n^2(.)$ according to (2).
5: **end for**
6: Update $\mu_N(.)$ according to (2).
7: **return** $\hat{x}_N^* = \text{argmax}_{x \in \mathcal{X}} \mu_N(x)$, where a tie is broken arbitrarily.
---

We note that MVR, similar to other standard GP bandit algorithms, requires optimizing an internal index created based on previous observations (here, $\sigma_{n-1}^2(.)$ and $\mu_N(.)$). Examples of other typical indices are UCB and EI, which are often referred to as *acquisition* functions. The index itself may be multi modal in general. It is however standard to assume a perfect optimization of the index, since the cost of evaluating $f$ is considered to dominate the cost of maximizing the index [19].

### 4.2 Regret Analysis

For the analysis of MVR, we assume there exists a fine discretization of the domain for RKHS elements, which is a standard assumption in the literature [see, e.g., 19, 20, 48].

**Assumption 4** *For each given $n \in \mathbb{N}$ and $f \in H_k$ with $\|f\|_{H_k} \leq B$, there exists a discretization $\mathcal{D}_n$ of $\mathcal{X}$ such that $f(x) - f([x]_n) \leq \frac{1}{\sqrt{n}}$, where $[x]_n = \text{argmin}_{x' \in \mathcal{D}_n} \|x' - x\|_{l^2}$ is the closest point in $\mathcal{D}_n$ to $x$, and $|\mathcal{D}_n| \leq CB^d n^{d/2}$, where $C$ is a constant independent of $n$ and $B$.*

Assumption 4 is a mild assumption that holds for typical kernels such as SE and Matérn [19, 20]. The following theorem provides a high probability bound on the regret performance of MVR, when the noise terms satisfy either Assumption 2 or 3.

**Theorem 3** *Consider the GP bandit problem, with a fixed $N$. Under Assumptions 1, 4, and (2 or 3), for $\delta \in (0,1)$, with probability at least $1 - \delta$, MVR satisfies*

$$r_N^{MVR} \leq \sqrt{\frac{2\gamma_N}{\log(1+\frac{1}{\lambda^2})N}}\left(2B + \beta(\frac{\delta}{3}) + \beta\left(\frac{\delta}{3C\left(B + \sqrt{N}\beta(2\delta/3N)\right)^d N^{d/2}}\right)\right) + \frac{2}{\sqrt{N}},$$

*where under Assumption 2, $\beta(\delta) = \frac{R}{\lambda}\sqrt{2\log(\frac{1}{\delta})}$, and under Assumption 3, $\beta(\delta) = \frac{1}{\lambda}\sqrt{2\left(\xi_0 \vee \frac{2\log(\frac{1}{\delta})}{h_0^2}\right)\log(\frac{1}{\delta})}$, and $B$, $R$, $h_0$, $\xi_0$ and $C$ are the constants specified in Assumptions 1, 2, 3 and 4.*

A detailed proof of the theorem is provided in Appendix D.

**Remark 2** *Under Assumptions 2 and 3, respectively, the regret bounds can be simplified as*

$$r_N^{MVR} = \mathcal{O}(\sqrt{\frac{\gamma_N \log(N^d/\delta)}{N}}), \quad and \quad r_N^{MVR} = \mathcal{O}\left(\sqrt{\frac{\gamma_N}{N}}\log(N^d/\delta)\right).$$

*For instance, in the case of Matérn-$\nu$ kernel, under Assumption 2 and 3, respectively,*

$$r_N^{MVR} = \mathcal{O}\left(N^{\frac{-\nu}{2\nu+d}}(\log(N))^{\frac{\nu}{2\nu+d}}\sqrt{\log(N^d/\delta)}\right), \quad and \ r_N^{MVR} = \mathcal{O}\left(N^{\frac{-\nu}{2\nu+d}}(\log(N))^{\frac{\nu}{2\nu+d}}\log(N^d/\delta)\right),$$

*which always converge to zero as $N$ grows.*

**Remark 3** *In the analysis of Theorem 3, we apply Assumption 4 to $\mu_N$ as well as $f$. For this purpose, we derive a high probability $B + \sqrt{N}\beta(2\delta/3N)$ upper bound on $\|\mu_N\|_{H_k}$ (see Lemma 4 in Appendix D), which appears in the regret bound expression.*

**Remark 4** *Theorem 3 holds for a fixed $N$. The result however easily extends to an* anytime *regret bound, using a standard probability union bound over $N$. Specifically, if we replace $\delta$ in the bound with $\delta_0 = \frac{6\delta}{\pi^2 N^2}$, the theorem holds, with probability $1-\delta$, for all $N \in \mathbb{N}$. Note that $\sum_{N=1}^{\infty}\frac{6\delta}{\pi^2 N^2} = \delta$.*

### 4.3 Optimal Order Simple Regret with SE and Matérn Kernels

To enable a direct comparison with the lower bounds on simple regret proven in [28, 29], in the following corollary, we state a dual form of Theorem 3 for the Matérn and SE kernels. Specifically we formalize the number of exploration trials required to achieve an average $\epsilon$ regret.

**Corollary 1** *Consider the GP bandit problem with an SE or a Matérn kernel. For $\epsilon \in (0,1)$, define $N_\epsilon = \min\{N \in \mathbb{N} : \mathbb{E}[r_n^{MVR}] \leq \epsilon, \forall n \geq N\}$. Under Assumptions 1, 4, and (2 or 3), upper bounds on $N_\epsilon$ are reported in Table 1.*

Table 1: The upper bounds on $N_\epsilon$ defined in Corollary 1 with SE or Matérn kernel.

| Kernel | Under Assumption 2 | Under Assumption 3 |
|---|---|---|
| SE | $N_\epsilon = \mathcal{O}\left((\frac{1}{\epsilon})^2 \log(\frac{1}{\epsilon})^{d+2}\right)$ | $N_\epsilon = \mathcal{O}\left((\frac{1}{\epsilon})^2 \log(\frac{1}{\epsilon})^{d+3}\right)$ |
| Matérn-$\nu$ | $N_\epsilon = \mathcal{O}\left((\frac{1}{\epsilon})^{2+\frac{d}{\nu}}(\log(\frac{1}{\epsilon}))^{\frac{4\nu+d}{2\nu}}\right)$ | $N_\epsilon = \mathcal{O}\left((\frac{1}{\epsilon})^{2+\frac{d}{\nu}}(\log(\frac{1}{\epsilon}))^{\frac{6\nu+2d}{2\nu}}\right)$ |

A proof is provided in Appendix E. [28, 29] showed that for the SE kernel, an average simple regret of $\epsilon$ requires $N_\epsilon = \Omega\left(\frac{1}{\epsilon^2}(\log(\frac{1}{\epsilon}))^{\frac{d}{2}}\right)$. For the Matérn-$\nu$ kernel they gave the analogous bound of $N_\epsilon = \Omega\left((\frac{1}{\epsilon})^{2+\frac{d}{\nu}}\right)$. They also reported significant gaps between these lower bounds and the existing results [see, e.g., 28, Table I]. Comparing with Corollary 1, our bounds are tight in all cases up to $\log(1/\epsilon)$ factors.

## 4.4 Comparing the Regret Bounds with Other Related Work

There are several Bayesian optimization algorithms namely GP-UCB [19], IGP-UCB, GP-TS [20], TruVar [70], GP-EI [50, 49] and KernelUCB [71] which enjoy theoretical upper bounds on regret under Assumptions 1, 2 and 4. The regret bounds of these algorithms grow at least as fast as $\mathcal{O}(\frac{\gamma_N}{\sqrt{N}})$. That does not necessarily converge to zero, since $\gamma_N$ can grow faster than $\sqrt{N}$, resulting in trivial regret bounds. For example, in the case of a Matérn-$\nu$ kernel, inserting $\gamma_N = \tilde{\mathcal{O}}(N^{\frac{d}{2\nu+d}})$ [22] gives an $\tilde{\mathcal{O}}(N^{\frac{d-2\nu}{4\nu+2d}})$ regret, which does not converge to zero for $d > 2\nu$, meaning the algorithm does not necessarily approach $f(x^*)$.

More recently, [21] developed a GP-UCB based algorithm, specific to Matérn family of kernels, that constructs a cover for the search space, as many hypercubes, and fits an independent GP to each cover element. This algorithm, referred to as $\pi$-GP-UCB, was proven to achieve diminishing regret for all $\nu > 1$ and $d \geq 1$. In particular, [21, Theorem 3] provided an upper bound on the regret of $\pi$-GP-UCB, which in our notation implies $r_N^{\pi\text{-GP-UCB}} = \tilde{\mathcal{O}}(N^{\frac{-2\nu-d}{4\nu+d(2d+4)}})$. In contrast, our bound for the case of Matérn kernel $r_N^{\text{MVR}} = \tilde{\mathcal{O}}(N^{\frac{-\nu}{2\nu+d}})$ is uniformly tighter that the bound on $r_N^{\pi\text{-GP-UCB}}$, for all $\nu > 1$ and $d \geq 1$. [72] introduced LP-GP-UCB where the GP model is augmented with local polynomial estimators to construct a multi-scale upper confidence bound guiding the sequential optimization. They further improved the regret bounds of [21] and showed that the performance of LP-GP-UCB matches the lower bound for some configuration of parameters $\nu$ and $d$ in the case of a Matérn kernel. Defining $\boldsymbol{I}_0 = (0, 1]$, $\boldsymbol{I}_1 = (1, \frac{d(d+1)}{2}]$, $\boldsymbol{I}_2 = (\frac{d(d+1)}{2}, \frac{d^2+5d+12}{4}]$ and $\boldsymbol{I}_3 = (0, \infty) \setminus \boldsymbol{I}_0 \cup \boldsymbol{I}_1 \cup \boldsymbol{I}_2$, their bounds on simple regret are as follows. For $\nu \in \boldsymbol{I}_0 \cup \boldsymbol{I}_1$, $r_N^{\text{LP-GP-UCB}} = \tilde{\mathcal{O}}(N^{\frac{-\nu}{2\nu+d}})$. For $\nu \in \boldsymbol{I}_2$, $r_N^{\text{LP-GP-UCB}} = \tilde{\mathcal{O}}(N^{\frac{-1}{2+d}})$. For $\nu \in \boldsymbol{I}_3$, $r_N^{\text{LP-GP-UCB}} = \tilde{\mathcal{O}}(N^{\frac{-4\nu+d(d+1)}{8\nu+2d(d+5)}})$ [see, 72, Sec. 3.2, for a detailed discussion on bounds on the simple regret of LP-GP-UCB]. In comparison, our bounds on simple regret match the $\Omega(N^{\frac{-\nu}{2\nu+d}})$ lower bound [28], up to logarithmic factors, with all parameters $\nu$ and $d$. In addition, LP-GP-UCB is impractical due to large constant factors, though a practical heuristic was also given.

Of important theoretical value, SupKernelUCB [71], which builds on episodic independent batches of observations, was proven to achieve $\tilde{\mathcal{O}}(\sqrt{\frac{\gamma_N}{N}})$ regret on a finite set ($|\mathcal{X}| < \infty$). SupKernelUCB is reported to perform poorly in practice [21, 47, 29]. While the proof of the regret bound for SupKernelUCB is given on a finite set, relying on the remarks from [21] and [29, Appendix A.4], we note that the bounds can be extended to a continuous domain using a discretization argument. That leads to $r_N^{\text{SupKernelUCB}} = \mathcal{O}(\log^2(N)\sqrt{\frac{\gamma_N}{N}})$, under similar assumptions as ours. In comparison, our bound on $r_N^{\text{MVR}}$ is tighter with a factor of $(\log(N))^{\frac{3}{2}}$.

It is noteworthy that our techniques do not directly apply to the analysis of cumulative regret of algorithms such as GP-UCB. The key difference is that in MVR the observation points $x_n$ are independent of the noise terms $\epsilon_n$ (although $x_n$ are allowed to depend on $\{x_j\}_{j<n}$, and $\hat{x}_N^*$ is allowed to depend on $\{x_n, \epsilon_n\}_{n \leq N}$), while in GP-UCB $x_n$ are allowed to depend on $\{\epsilon_j\}_{j<n}$ (see also Sec. 3.3). It remains an interesting open question whether the existing analysis of the regret performance of GP-UCB [20] is tight or the gap with the lower bound [28] is an artifact of its proof. For more details on this open question, see [73].

# 5 Experiments

In this section, we provide numerical experiments on the simple regret performance of MVR, Improved GP-UCB (IGP-UCB) as presented in [20], and GP-PI and GP-EI as presented in [26].

We follow the experiment set up in [20] to generate test functions from the RKHS. First, 100 points are uniformly sampled from interval $[0, 1]$. A GP sample with kernel $k$ is drawn over these points. Given this sample, the mean of posterior distribution is used as the test function $f$. Parameter $\lambda^2$ is set to 1% of the function range. For IGP-UCB we set the parameters exactly as described in [20]. The GP model is equipped with SE or Matérn-2.5 kernel with $l = 0.2$. We use 2 different models for the noise: a zero mean Gaussian with variance equal to $\lambda^2$ (a sub-Gaussian distribution) and a zero mean Laplace with scale parameter equal to $\lambda$ (a light-tailed distribution). We run each experiment over 25 independent trials and plot the average simple regret in Figure 1. More experiments on

two commonly used benchmark functions for Bayesian optimization (Rosenbrock and Hartman3) are reported in Appendix F. Further details on the experiments are provided in the supplementary material.

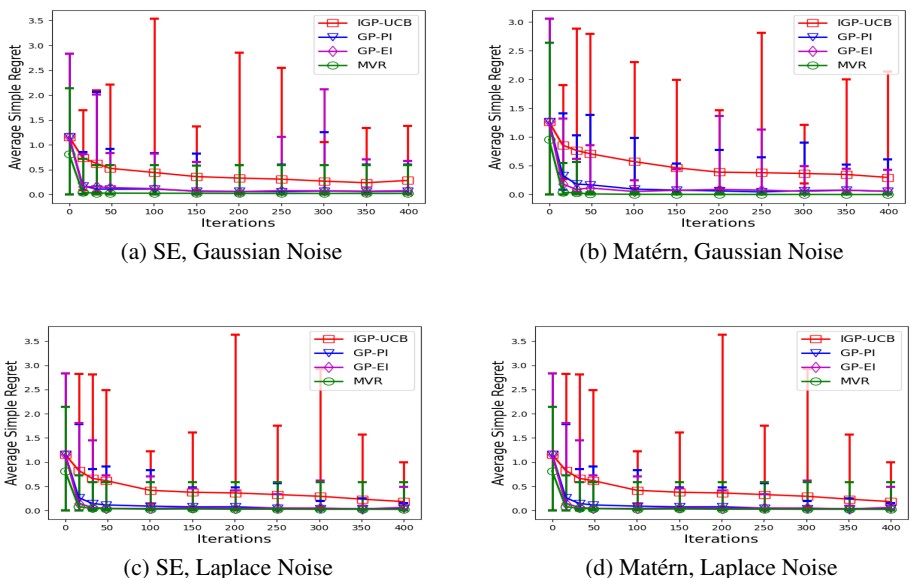

Figure 1: Comparison of the simple regret performance of Bayesian optimization algorithms on samples from RKHS.

(a) SE, Gaussian Noise

(b) Matérn, Gaussian Noise

(c) SE, Laplace Noise

(d) Matérn, Laplace Noise

## 6 Discussion

In this paper, we proved novel and sharp confidence intervals for GP models applicable to RKHS elements. We then built on these results to prove $\tilde{\mathcal{O}}(\sqrt{\gamma_N/N})$ bounds for the simple regret of an adaptive exploration algorithm under the framework of GP bandits. In addition, for the practically relevant SE and Matérn kernels, where a lower bound on regret is known [28, 29], we showed the order optimality of our results up to logarithmic factors. That closes a significant gap in the literature of analysis of Bayesian optimization algorithms under the performance measure of simple regret.

The limitation of our work adhering to simple regret is that neither our theoretical nor experimental results proves that MVR is necessarily the right algorithm in practice. Overall, exploration-exploitation oriented algorithms such as GP-UCB may perform worse than MVR in terms of simple regret due to two reasons. One is over-exploitation of local maxima when $f$ is multimodal, and the other is dependence on an exploration-exploitation balancing hyper-parameter that is often set too conservatively, to guarantee low regret bounds. Furthermore, their existing analytical regret bounds are suboptimal and possibly trivial (non-diminishing; when $\gamma_N$ grows faster than $\sqrt{N}$, as discussed). On the other hand, when compared in terms of *cumulative* regret ($\sum_{n=1}^{N} f(x^*) - f(x_n)$), MVR suffers from a linear regret.

The main value of our work is in proving tight bounds on the simple regret of a GP based exploration algorithm, when other Bayesian optimization algorithms such as GP-UCB lack a proof for an always diminishing regret under the same setting as ours. It remains an open question whether the possibly trivial regret bounds of GP-UCB (as well as GP-TS and GP-EI, whose analysis is inspired by that of GP-UCB) is a fundamental limitation or an artifact of the proof [73].

## Funding Disclosure

This work was funded by MediaTek Research.

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
