# Supplementary Material: "Optimal Order Simple Regret for Gaussian Process Bandits"

**Sattar Vakili**[*]**, Nacime Bouziani**[+]**, Sepehr Jalali**[*]**, Alberto Bernacchia**[*]**, Da-shan Shiu**[*]

[*] MediaTek Research

{sattar.vakili, sepehr.jalali, alberto.bernacchia, ds.shiu}@mtkresearch.com

[+]Imperial College London

n.bouziani18@imperial.ac.uk

## A   Constructive Definition of RKHS

A constructive definition of RKHS requires the use of Mercer theorem, which provides an alternative representation for kernels as an inner product of infinite dimensional feature maps [see, e.g., 1, Theorem 4.1].

**Mercer Theorem:**   Let $k$ be a continuous kernel with respect to a finite Borel measure. There exists $\{(\lambda_m, \phi_m)\}_{m=1}^{\infty}$ such that $\lambda_m \in \mathbb{R}^+$, $\phi_m \in H_k$, for $m \geq 1$, and

$$k(x, x') = \sum_{m=1}^{\infty} \lambda_m \phi_m(x) \phi_m(x').$$

The RKHS can consequently be represented in terms of $\{(\lambda_m, \phi_m)\}_{m=1}^{\infty}$ using Mercer's representation theorem [see, e.g., 1, Theorem 4.2].

**Mercer's Representation Theorem:**   Let $\{(\lambda_m, \phi_m)\}_{m=1}^{\infty}$ be the same as in Mercer Theorem. Then, the RKHS of $k$ is given by

$$H_k = \left\{ f(\cdot) = \sum_{m=1}^{\infty} w_m \lambda_m^{\frac{1}{2}} \phi_m(\cdot) : ||f||_{H_k}^2 \triangleq \sum_{m=1}^{\infty} w_m^2 < \infty \right\}.$$

Mercer's representation theorem indicates that $\{\lambda_m^{\frac{1}{2}} \phi_m\}_{m=1}^{\infty}$ form an orthonormal basis for $H_k$. It also provides a constructive definition for the RKHS as the span of this orthonormal basis, and a definition for the norm of a member $f$ as the $l_2$ norm of the weights $w_m$.

The RKHS of Matérn is equivalent to a Sobolev space with parameter $\nu + \frac{d}{2}$ [1, 2]. This observation provides an intuitive interpretation for the norm of Matérn RKHS as proportional to the cumulative $L_2$ norm of the *weak derivatives* of $f$ up to $\nu + \frac{d}{2}$ order. I.e., in the case of Matérn family, Assumption 1 on the norm of $f$ translates to the existence of weak derivatives of $f$ up to $\nu + \frac{d}{2}$ order, which can be understood as a versatile measure for the smoothness of $f$ controlled by $\nu$. In the case of SE kernel, the regularity assumption implies the existence of all weak derivatives of $f$. For the details on the definition of weak derivatives and Sobolev spaces see [3].

## B   Proof of Proposition 1

Recall the notations $Y_n = [y_1, y_2, \ldots, y_n]^{\top}$, $F_n = [f(x_1), f(x_2), \ldots, f(x_n)]^{\top}$, $Z_n^{\top}(x) = k^{\top}(x, X_n) \left( k(X_n, X_n) + \lambda^2 I_n \right)^{-1}$. Let $\zeta_i(x) = [Z_n(x)]_i$. From the closed form expression for the posterior mean of GP models, we have $\mu_n(x) = Z_n^{\top}(x) Y_n$.

The proof of Proposition 1 uses the following lemma.

**Lemma 1** *For a positive definite kernel $k$ and its corresponding RKHS, the following holds.*

$$\sup_{f:||f||_{H_k}\leq 1}\left(f(x)-\sum_{i=1}^{n}\zeta_i(x)f(x_i)\right)^2=\left|\left|k(.,x)-\sum_{i=1}^{n}\zeta_i(x)k(.,x_i)\right|\right|_{H_k}^2. \tag{1}$$

The lemma establishes the equivalence of the RKHS norm of a linear combination of feature vectors induced by $k$ to the supremum of the linear combination of the corresponding function values, over the functions in the unit ball of the RKHS. For a proof, see [1, Lemma 3.9].

Expanding the RKHS norm in the right hand side through an algebraic manipulation, we get

$$\left|\left|k(.,x)-\sum_{i=1}^{n}\zeta_i(x)k(.,x_i)\right|\right|_{H_k}^2$$

$$= k(x,x)-2\sum_{i=1}^{n}\zeta_i(x)k(x,x_i)+\sum_{i=1}^{n}\sum_{j=1}^{n}\zeta_i(x)\zeta_j(x)k(x_i,x_j)$$

$$= k(x,x)-2\sum_{i=1}^{n}\zeta_i(x)k(x,x_i)+(Z_n(x))^\top k(X_n,X_n)Z_n(x)$$

$$= k(x,x)-2(k(x,X_n))^\top(k(X_n,X_n)+\lambda^2 I_n)^{-1}k(x,X_n)$$
$$\quad + (k(x,X_n))^\top(k(X_n,X_n)+\lambda^2 I_n)^{-1}k(X_n,X_n)(k(X_n,X_n)+\lambda^2 I_n)^{-1}k(x,X_n)$$

$$= k(x,x)-2k(x,X_n)^\top(k(X_n,X_n)+\lambda^2 I_n)^{-1}k(x,X_n)$$
$$\quad + k(x,X_n)^\top(k(X_n,X_n)+\lambda^2 I_n)^{-1}(k(X_n,X_n)+\lambda^2 I_n-\lambda^2 I_n)(k(X_n,X_n)+\lambda^2 I_n)^{-1}k(x,X_n)$$

$$= k(x,x)-2k(x,X_n)^\top(k(X_n,X_n)+\lambda^2 I_n)^{-1}k(x,X_n)$$
$$\quad + k(x,X_n)^\top(k(X_n,X_n)+\lambda^2 I_n)^{-1}k(x,X_n)-\lambda^2 k(x,X_n)^\top(k_{X_n,X_n}+\lambda^2 I_n)^{-2}k(x,X_n)$$

$$= k(x,x)-(k(X_n,X_n))^\top(k(X_n,X_n)+\lambda^2 I_n)^{-1}k(X_n,X_n)-\lambda^2 k(x,X_n)^\top(k(X_n,X_n)+\lambda^2 I_n)^{-2}k(x,X_n)^\top$$

$$= \sigma_n^2(x)-\lambda^2(Z_n(x))^\top Z_n(x)$$

$$= \sigma_n^2(x)-\lambda^2\left|\left|Z_n(x)\right|\right|^2.$$

The first equation uses the reproducing property of the RKHS. The second equation results from expressing the series in the vector product form. The third equation follows from the definition of $Z_n(x)$. The fourth and fifth equations follow from adding and subtracting a $\lambda^2 I_n$ term to the covariance matrix and some algebraic calculation. Sixth equation uses the closed form expression for the posterior variance of GP models and the definition of $Z_n(x)$.

Rearranging and combining with (1), we arrive at

$$\sigma_n^2(x)=\sup_{f:||f||_{H_k}\leq 1}\left(f(x)-Z_n^\top(x)F_n\right)^2+\lambda^2\left|\left|Z_n(x)\right|\right|^2.$$

## C   Proof of Theorems 1 and 2

Recall the closed form expression for the posterior mean of GP models $\mu_n(x)=Z_n^\top(x)Y_n$. We can expand the prediction error in terms of prediction error due to noise-free observations and the effect of noise as follows

$$f(x)-\mu_n(x) = f(x)-Z_n^\top(x)Y_n$$
$$= f(x)-Z_n^\top(x)F_n-Z_n^\top(x)E_n. \tag{2}$$

We now use Proposition 1 to bound both terms.

**Prediction error due to noise free observations** can be simply bounded by $B\sigma_n$ as a direct result of Proposition 1. Specifically let $\tilde{f}(.) = f(.)/B$ so that $||\tilde{f}||_{H_k} \leq 1$. Also, let $\tilde{F}_n = [\tilde{f}(x_1), \tilde{f}(x_2), \ldots, \tilde{f}(x_n)]^\top$.

$$
\begin{aligned}
|f(x) - Z_n^\top(x)F_n| &= B|\tilde{f}(x) - Z_n^\top(x)\tilde{F}_n| \\
&\leq B\sigma_n(x),
\end{aligned}
\tag{3}
$$

where the inequality follows from Proposition 1 and $||\tilde{f}||_{H_k} \leq 1$.

We now proceed using Assumption 2 to prove Theorem 1.

**The effect of noise** is bounded using the sub-Gaussianity assumption. In particular, we show that $Z_n^\top(x)E_n$ is a sub-Gaussian random variable whose moment generating function is bounded by that of a Gaussian random variable with variance $\frac{R^2\sigma_n^2(x)}{\lambda^2}$.

$$
\begin{aligned}
\mathbb{E}\left[\exp(Z_n^\top(x)E_n)\right] &= \mathbb{E}\left[\exp\left(\sum_{i=1}^{n}\zeta_i(x)\epsilon_i\right)\right] \\
&= \prod_{i=1}^{n}\mathbb{E}[\exp(\zeta_i(x)\epsilon_i)] \\
&\leq \prod_{i=1}^{n}\exp(\frac{R^2(\zeta_i(x))^2}{2}) \\
&= \exp\left(\frac{R^2\sum_{i=1}^{n}(\zeta_i(x))^2}{2}\right) \\
&= \exp\left(\frac{R^2||Z_n(x)||^2}{2}\right) \\
&\leq \exp\left(\frac{R^2\sigma_n^2(x)}{2\lambda^2}\right).
\end{aligned}
$$

where the second equation is a result of independence of $\zeta_i(x)\epsilon_i$ terms that follows from the assumptions of i.i.d. noise terms and $X_n$ being independent of $E_n$. The first inequality holds by Assumption 2. We utilize Proposition 1 to conclude that $||Z_n(x)||^2 \leq \frac{\sigma_n^2(x)}{\lambda^2}$ which results in the second inequality. Thus, using Chernoff-Hoeffding inequality [4],

$$
\begin{aligned}
Z_n(x)E_n &\geq -\frac{\sigma_n(x)R}{\lambda}\sqrt{2\log(\frac{1}{\delta})} \quad \text{w.p. at least } 1-\delta, \\
Z_n(x)E_n &\leq \frac{\sigma_n(x)R}{\lambda}\sqrt{2\log(\frac{1}{\delta})} \quad \text{w.p. at least } 1-\delta.
\end{aligned}
\tag{4}
$$

Putting together (2), (3) and (4), Theorem 1 is proven.

We now move to the proof of Theorem 2. For the simplicity of the notation let us use

$$
\tau = ||Z_n(x)||\sqrt{2(\xi_0 \vee \frac{2\log(1/\delta)}{h_0^2})\log(\frac{1}{\delta})},
\tag{5}
$$

$$
\xi = \xi_0 \vee \frac{2\log(1/\delta)}{h_0^2}.
\tag{6}
$$

We have, for $\theta = \frac{\tau}{\xi||Z_n(x)||^2}$,

$$
\begin{aligned}
\Pr[Z_n^\top(x)E_n \geq \tau] &= \Pr\left[\exp(\theta Z_n^\top(x)E_n) \geq \exp(\theta\tau)\right] \\
&\leq \exp(-\theta\tau)\mathbb{E}\left[\exp(\theta Z_n^\top(x)E_n)\right] \\
&= \exp(-\theta\tau)\mathbb{E}\left[\exp\left(\sum_{i=1}^n \theta\zeta_i(x)\epsilon_i\right)\right] \\
&= \exp(-\theta\tau)\prod_{i=1}^n \mathbb{E}\left[\exp(\theta\zeta_i(x)\epsilon_i)\right] \\
&\leq \exp(-\theta\tau)\prod_{i=1}^n \exp\left(\frac{1}{2}\xi_0\theta^2(\zeta_i(x))^2\right) \\
&= \exp\left(\frac{1}{2}\xi_0\theta^2||Z_n(x)||^2 - \theta\tau\right) \\
&= \exp\left(\frac{\xi_0\tau^2}{2\xi^2||Z_n(x)||^2} - \frac{\tau^2}{\xi||Z_n(x)||^2}\right) \\
&\leq \exp(-\frac{\tau^2}{2\xi||Z_n(x)||^2}) \\
&= \delta.
\end{aligned}
\tag{7}
$$

The first line is obtained since $\exp(\theta z)$ in an increasing function in $z$. The first inequality amounts for an application of Markov inequality. The fourth line is a result of independence of $\zeta_i(x)\epsilon_i$ terms that follows from the assumptions of i.i.d. noise terms and $X_n$ being independent of $E_n$. The second inequality holds by definition of light-tailed distributions. Notice that the careful choice of $\tau$ and $\theta$ ensures $\theta\zeta_i(x) \leq h_0$, which will be validated next. The seventh line is obtained by replacing the value of $\theta$. The last inequality is obtained by $\xi_0 \leq \xi$. The last line is resulted from replacing the value of $\tau$ from (5).

It remains to validate $\theta\zeta_i(x) \leq h_0$.

$$
\begin{aligned}
\theta\zeta_i(x) &= \frac{\tau}{\xi||Z_n(x)||^2}\zeta_i(x) \\
&= \frac{\sqrt{2\log(\frac{1}{\delta})}\zeta_i(x)}{\sqrt{\xi}||Z_n(x)||} \\
&\leq h_0\frac{\zeta_i(x)}{||Z_n(x)||} \\
&\leq h_0,
\end{aligned}
$$

where we replace $\theta = \frac{\tau}{\xi||Z_n||^2}$, and the values of $\tau$ and $\xi$ from (5) and (6), respectively. For the first inequality, notice that $\frac{2\log(1/\delta)}{h_0^2} \leq \xi$ from the definition of $\xi$ (6). For the second inequality notice that $\zeta_i(x) \leq ||Z_n(x)||$.

We thus proved

$$
Z_n(x)E_n \leq \tau, \quad \text{w.p. at least } 1 - \delta. \tag{8}
$$

Similarly, we can prove

$$
Z_n(x)E_n \geq -\tau, \quad \text{w.p. at least } 1 - \delta. \tag{9}
$$

Replacing $||Z_n(z)|| \leq \frac{R}{\lambda}\sigma_n(x)$ from Proposition 1 in the value of $\tau$ (5), and combining (8) and (9) with (2) and (3), Theorem 2 is proven.

# D   Proof of Theorem 3

The MVR algorithm selects the points with the highest variance first. Thus, $\forall x \in \mathcal{X}$,

$$\sigma_{n-1}^2(x) \leq \sigma_{n-1}^2(x_n). \tag{10}$$

By definition of conditional variance of normal distributions and due to positive definiteness of covariance matrix, conditioning on a larger set of points reduces the variance. Specifically, we have, for all $x \in \mathcal{X}$ and $\forall n \leq N$,

$$\sigma_N^2(x) \leq \sigma_{n-1}^2(x). \tag{11}$$

Combining (10) and (11), we have, $\forall x \in \mathcal{X}$ and $\forall n \leq N$,

$$\sigma_N^2(x) \leq \sigma_{n-1}^2(x_n).$$

Averaging both sides over $n$ (from 1 to $N$), we have

$$\sigma_N^2(x) \leq \frac{1}{N} \sum_{n=1}^N \sigma_{n-1}^2(x_n). \tag{12}$$

We now use the following lemma to bound $\sigma_N^2(x)$.

**Lemma 2** *Recall $\mathcal{I}(Y_n; \hat{f}) = \frac{1}{2} \log \det(I_n + \frac{1}{\lambda^2} k(X_n, X_n))$. For the cumulative conditional variance of the GP model, we have*

$$\sum_{n=1}^N \sigma_{n-1}^2(x_n) \leq \frac{2}{\log(1 + \frac{1}{\lambda^2})} \mathcal{I}(Y_N; \hat{f}).$$

A proof can be found in [5].

We thus have, for all $x \in \mathcal{X}$,

$$\begin{aligned}
\sigma_N^2(x) &\leq \frac{2\mathcal{I}(Y_n; \hat{f})}{\log(1 + \frac{1}{\lambda^2})N} \\
&\leq \frac{2\gamma_N}{\log(1 + \frac{1}{\lambda^2})N},
\end{aligned} \tag{13}$$

where $\gamma_N$ is the maximal information gain defined in Sec. 2.4.

Let $B_0(\delta) = B + \sqrt{N}\beta(2\delta/N)$. At the end of this section, in Lemma 4, we prove that

$$\|\mu_N\|_{H_k} \leq B_0(\delta), \text{ w.p. at least } 1 - \delta. \tag{14}$$

Notice that $\mu_n$ is a random function due to the randomness in noise. Let us define the event $\mathcal{E} = \{\|\mu_N\|_{H_k} \leq B_0(\delta/3)\}$. We have $\Pr[\mathcal{E}] \geq 1 - \frac{\delta}{3}$.

Under event $\mathcal{E}$, we use Assumption 4 on the existence of a discretization $\mathcal{D}_N(\delta)$ of $\mathcal{X}$ such that $f(x) - f([x]_N) \leq \frac{1}{\sqrt{N}}$, $\mu_N(x) - \mu_N([x]_N) \leq \frac{1}{\sqrt{N}}$, and $|\mathcal{D}_N(\delta)| \leq CB_0^d(\delta/3)N^{d/2}$. Notice that we do not need to actually create this discretization. We only use its existence to handle the analysis in a continuous space, using a probability union bound based on this discretization.

For a fixed $x \in \mathcal{D}_N$, from the confidence bounds for GP models proven in Theorems 1 and 2, we have

$$f(x) \geq \mu_n(x) - (B + \beta(\frac{\delta}{3|\mathcal{D}_N(\delta)|}))\sigma_n(x), \text{ w.p. at least } 1 - \frac{\delta}{3|\mathcal{D}_N(\delta)|}.$$

Using a probability union bound, we have, $\forall x \in \mathcal{D}_N(\delta)$

$$f(x) \geq \mu_n(x) - (B + \beta(\frac{\delta}{3|\mathcal{D}_N(\delta)|}))\sigma_n(x), \text{ w.p. at least } 1 - \frac{\delta}{3}. \tag{15}$$

We thus have, under event $\mathcal{E}$,

$$
\begin{aligned}
f(x^*) - f(\hat{x}_N^*) &= f(x^*) - f([\hat{x}_N^*]_N) + f([\hat{x}_N^*]_N) - f(\hat{x}_N^*) \\
&\leq f(x^*) - f([\hat{x}_N^*]_N) + \frac{1}{\sqrt{N}} \\
&\leq f(x^*) - \mu_N(x^*) + \mu_N(\hat{x}_N^*) - f([\hat{x}_N^*]_N) + \frac{1}{\sqrt{N}} \\
&= f(x^*) - \mu_N(x^*) + \mu_N(\hat{x}_N^*) - \mu_N([\hat{x}_N^*]_N) + \mu_N([\hat{x}_N^*]_N) - f([\hat{x}_N^*]_N) + \frac{1}{\sqrt{N}} \\
&\leq f(x^*) - \mu_N(x^*) + \mu_N([\hat{x}_N^*]_N) - f([\hat{x}_N^*]_N) + \frac{2}{\sqrt{N}}.
\end{aligned}
$$

The first inequality comes from Assumption 4 on discretization $\mathcal{D}_N(\delta)$ and $f$. The second inequality comes from the definition of MVR which ensures $\mu_N(\hat{x}_N^*) \geq \mu_N(x)$, for all $x \in \mathcal{X}$. For the last inequality, we use Assumption 4 on discretization $\mathcal{D}_N(\delta)$ and $\mu_N$. Notice that under event $\mathcal{E}$, the posterior mean of the GP model belongs to the same RKHS with its norm bounded by $B_0(\delta/3)$.

Thus, noting that the inequality given in (15), the confidence interval for $f(x^*)$ with $1 - \delta/3$ confidence, and $\mathcal{E}$, each hold true with probability at least $1 - \frac{\delta}{3}$, using a probability union bound, we have

$$
\begin{aligned}
f(x^*) - f(\hat{x}_N^*) &\leq (B + \beta(\frac{\delta}{3}))\sigma_N(x^*) + (B + \beta(\frac{\delta}{3|\mathcal{D}_N(\delta)|}))\sigma_N([\hat{x}_N^*]_N) \\
&\quad + \frac{2}{\sqrt{N}}, \quad \text{w.p. at least } 1 - \delta.
\end{aligned} \tag{16}
$$

Using (13) to bound $\sigma_N(x^*)$ and $\sigma_N([\hat{x}_N^*]_N)$, and replacing $|\mathcal{D}_N(\delta)|$ with its upper bound, we get

$$
\begin{aligned}
f(x^*) - f(\hat{x}_N^*) &\leq \sqrt{\frac{2\gamma_N}{\log(1 + \frac{1}{\lambda^2})N}} \left( 2B + \beta(\frac{\delta}{3}) + \beta(\frac{\delta}{3C(B + \sqrt{N}\beta(2\delta/3N))^d N^{d/2}}) \right) \\
&\quad + \frac{2}{\sqrt{N}}, \text{ w.p. at least } 1 - \delta,
\end{aligned} \tag{17}
$$

which completes the proof.

We now prove a high probability upper bound on $\|\mu_n\|_{H_k}$.

Let us first formally state the equivalence of the posterior mean in GP models and the regressor in kernel ridge regression.

**Lemma 3** *Conditioned on a set of noisy observation $\mathcal{H}_n$ from $f$, recall the expression for the posterior mean of the GP model $\mu_n(x) = Z_n^\top(x)Y_n$. We have the following equality*

$$
\mu_n = argmin_{g \in H_k} \left( \lambda^2 \|g\|_{H_k}^2 + \sum_{i=1}^n (g(x_i) - y_i)^2 \right). \tag{18}
$$

For a proof, see [1, Theorem 3.4]. Lemma 3 establishes the equivalence of the posterior mean in GP models and the regressor in kernel ridge regression. It indicates that the posterior mean of GP models is a mean squared error estimator, regularized by the RKHS norm, where $\lambda^2$ is the regularization parameter. We use this lemma to show that the posterior mean of the GP model, with high probability, lives in the same RKHS as $f$.

**Lemma 4** *Conditioned on a set of noisy observation $\mathcal{H}_n$ from $f$ with $\|f\|_{H_k} \leq B$, the RKHS norm of the posterior mean of the GP model $\mu_n(x) = Z_n^\top(x)Y_n$ satisfies the following*

$$
\|\mu_n\|_{H_k} \leq B + \sqrt{n}\beta(2\delta/n), \text{ w.p. at least } 1 - \delta, \tag{19}
$$

*where $\beta(\delta) = \frac{R}{\lambda}\sqrt{2\log(\frac{1}{\delta})}$ under Assumption 2, and $\beta(\delta) = \frac{1}{\lambda}\sqrt{2(\xi_0 \vee \frac{2\log(1/\delta)}{h_0^2})\log(\frac{1}{\delta})}$ under Assumption 3.*

**Proof of Lemma 4:** We have

$$
\begin{aligned}
\|\mu_n\|_{H_k} &= \|Z_n^\top(.)F_n + Z_n^\top(.)E_n\|_{H_k} \\
&\leq \|Z_n^\top(.)F_n\|_{H_k} + \|Z_n^\top(.)E_n\|_{H_k}.
\end{aligned}
\tag{20}
$$

From Lemma 3, we have

$$
\lambda^2\|Z_n^\top(.)F_n\|_{H_k}^2 + \sum_{i=1}^n (Z_n^\top(x_i)F_n - f(x_i))^2 \leq \lambda^2\|f\|_{H_k}^2 + \sum_{i=1}^n (f(x_i) - f(x_i))^2
$$

Thus,

$$
\|Z_n^\top(.)F_n\|_{H_k} \leq \|f\|_{H_k},
\tag{21}
$$

where $\|f\|_{H_k} \leq B$. It thus remains to bound the second term on the right hand side of (20).

$$
\begin{aligned}
\|Z_n^\top(.)E_n\|_{H_k}^2 &= \|k^\top(x, X_n)\left(k(X_n, X_n) + \lambda^2 I_n\right)^{-1} E_n\|_{H_k}^2 \\
&= E_n^\top\left(k(X_n, X_n) + \lambda^2 I_n\right)^{-1} k(X_n, X_n)\left(k(X_n, X_n) + \lambda^2 I_n\right)^{-1} E_n \\
&= E_n^\top\left(k(X_n, X_n) + \lambda^2 I_n\right)^{-1}\left(k(X_n, X_n) + \lambda^2 I_n\right)\left(k(X_n, X_n) + \lambda^2 I_n\right)^{-1} E_n \\
&\quad - \lambda^2 E_n^\top\left(k(X_n, X_n) + \lambda^2 I_n\right)^{-2} E_n \\
&\leq E_n^\top\left(k(X_n, X_n) + \lambda^2 I_n\right)^{-1} E_n \\
&\leq \frac{1}{\lambda^2}\|E_n\|_{l_2}^2,
\end{aligned}
$$

where for the second line we used the reproducing property of the RKHS, for the first inequality we used positive definiteness of $\left(k(X_n, X_n) + \lambda^2 I_n\right)^{-2}$ that is a result of positive definiteness of $k(X_n, X_n)$, and for the last inequality we used positive definiteness of $k(X_n, X_n)$.

Under Assumption 2, as a result of Chernoff-Hoeffding inequality,

$$
\epsilon_i^2 \leq 2R^2 \log(\frac{1}{2\delta'}), \text{ w.p. at least } 1 - \delta'.
$$

Using a probability union bound over $i = 1, 2, \ldots, n$, with $\delta' = \frac{\delta}{n}$,

$$
\frac{1}{\lambda^2}\|E_n\|_{l_2}^2 \leq \frac{2nR^2}{\lambda^2}\log(\frac{n}{2\delta}), \text{ w.p. at least } 1 - \delta.
\tag{22}
$$

Under Assumption 3, as a result of (7) (with $n = 1$, and $Z_n = 1$),

$$
\epsilon_i^2 \leq 2(\xi_0 \vee \frac{2\log(1/2\delta')}{h_0^2})\log(\frac{1}{2\delta'}), \text{ w.p. at least } 1 - \delta'.
$$

Using a probability union bound over $i = 1, 2, \ldots, n$, with $\delta' = \frac{\delta}{n}$,

$$
\frac{1}{\lambda^2}\|E_n\|_{l_2}^2 \leq \frac{2n}{\lambda^2}(\xi_0 \vee \frac{2\log(n/2\delta)}{h_0^2})\log(\frac{n}{2\delta}), \text{ w.p. at least } 1 - \delta.
\tag{23}
$$

Combining the bounds on the both terms on the right hand side of (20), the lemma is proven.

## E    Proof of Corollary 1

We use Theorem 3 to derive a bound on the expected regret of MVR.

First, notice that $|f(x)| \leq k_0 B$ where $k_0^2 = \max_{x \in \mathcal{X}} k(x, x)$, which can be proven using the reproducing property of the RKHS.

$$
\begin{aligned}
|f(x)| &= |\langle f(.), k(., x)\rangle_{H_k}| \\
&\leq \|f\|_{H_k}\|k(., x)\|_{H_k} \\
&= \|f\|_{H_k}\sqrt{k(x, x)} \\
&\leq k_0 B.
\end{aligned}
$$

So, we have $\max_{x \in \mathcal{X}} f(x^*) - f(x) \leq 2k_0 B$. Let $\mathcal{E}$ denote the even that $r_N^{\text{MVR}} \leq \bar{r}$, where

$$\bar{r} = \sqrt{\frac{2\gamma_N}{\log(1 + \frac{1}{\lambda^2})N}} \left( 2B + \beta(\frac{1}{3\sqrt{N}}) + \beta \left( \frac{1}{3C\sqrt{N} \left(B + \sqrt{N}\beta(2/3N\sqrt{N})\right)^d N^{d/2}} \right) \right) + \frac{2}{\sqrt{N}}$$

is the upper bound on regret given in Theorem 3 with $\delta = \frac{1}{\sqrt{N}}$. From Theorem 3, we have $\Pr[\mathcal{E}] \geq 1 - \frac{1}{\sqrt{N}}$.

Using the law of total expectation, we have

$$
\begin{aligned}
\mathbb{E}[r_N^{\text{MVR}}] &= \mathbb{E}\left[r_N^{\text{MVR}} | \mathcal{E}\right] \Pr[\mathcal{E}] + \mathbb{E}\left[r_N^{\text{MVR}} | \bar{\mathcal{E}}\right] \Pr[\bar{\mathcal{E}}] \\
&\leq \bar{r} + \frac{2k_0 B}{\sqrt{N}} \\
&= \mathcal{O}\left( \sqrt{\frac{\gamma_N}{N}} \beta(\frac{1}{N^{d+\frac{1}{2}}}) \right).
\end{aligned}
$$

Under Assumption 2,

$$\mathbb{E}[r_N^{\text{MVR}}] = \mathcal{O}\left( \sqrt{\frac{\gamma_N}{N} \log(N^{d+\frac{1}{2}})} \right). \tag{24}$$

Under Assumption 3,

$$\mathbb{E}[r_N^{\text{MVR}}] = \mathcal{O}\left( \sqrt{\frac{\gamma_N}{N}} \log(N^{d+\frac{1}{2}}) \right). \tag{25}$$

In the case of SE kernel, $\gamma_N = \mathcal{O}\left((\log(N))^{d+1}\right)$ [5]. Selecting $N \propto (\frac{1}{\epsilon})^2 (\log(\frac{1}{\epsilon}))^{d+2}$ and $N \propto (\frac{1}{\epsilon})^2 (\log(\frac{1}{\epsilon}))^{d+3}$, with proper constants, under Assumptions 2 and 3, respectively, results in $\mathbb{E}[r_N^{\text{MVR}}] \leq \epsilon$.

In the case of Matérn kernel, $\gamma_N = \mathcal{O}\left( N^{\frac{d}{2\nu+d}} (\log(N))^{\frac{2\nu}{2\nu+d}} \right)$ [6]. Selecting $N \propto (\frac{1}{\epsilon})^{2+\frac{d}{\nu}} (\log(\frac{1}{\epsilon}))^{\frac{4\nu+d}{2\nu}}$ and $N \propto (\frac{1}{\epsilon})^{2+\frac{d}{\nu}} (\log(\frac{1}{\epsilon}))^{\frac{6\nu+2d}{2\nu}}$, with proper constants, under Assumptions 2 and 3, respectively, results in $\mathbb{E}[r_N^{\text{MVR}}] \leq \epsilon$.

Finding the exact constants requires solving a non-linear equation involving $\log$ function which is a tedious task.

Noticing that $\mathbb{E}[r_n^{\text{MVR}}]$ is a decreasing function in $n$ completes the proof.

# F  Supplemental Material on the Experiments

In this section, we provide further details on the experiments and the source code. We also provide additional experiments on two commonly used benchmark functions for Bayesian optimization.

## F.1  Additional Experiments

In Section 5, we provided experiments on comparison of the simple regret performance of Bayesian optimization algorithms on synthetically generated functions in RKHS. In this section, we consider two commonly used benchmark functions for Bayesian optimization: Hartman3 and Rosenbrock as presented in [7, 8]. The parameters of the kernels, noise and $\lambda$ are set exactly as described in Section 5. We plot the average simple regret for all four learning algorithms considered in Section 5, over 50 independent experiments, with Hartman3 test function in Figure 1, and with Rosenbrock test function in Figure 2. The details on the source code is provided in the next section. The data used for generating the figures is provided in the supplementary material.

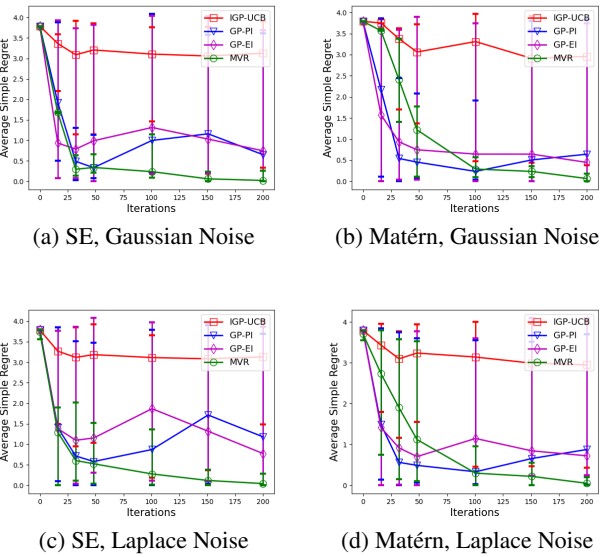

(a) SE, Gaussian Noise          (b) Matérn, Gaussian Noise

(c) SE, Laplace Noise          (d) Matérn, Laplace Noise

Figure 1: Comparison of the simple regret performance of Bayesian optimization algorithms on Hartman3 test function.

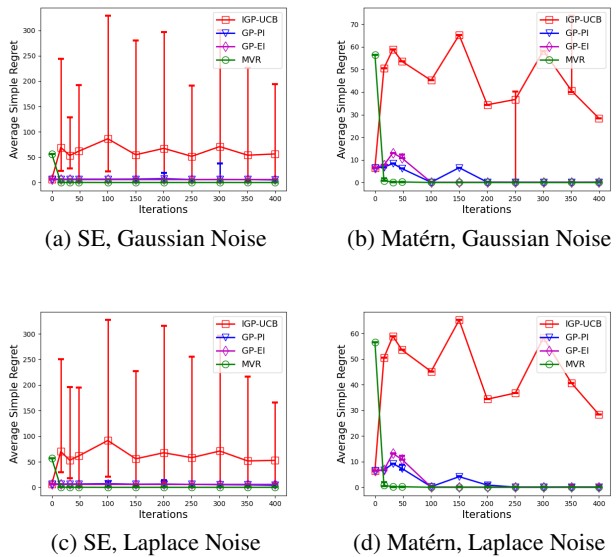

(a) SE, Gaussian Noise          (b) Matérn, Gaussian Noise

(c) SE, Laplace Noise          (d) Matérn, Laplace Noise

Figure 2: Comparison of the simple regret performance of Bayesian optimization algorithms on Rosenbrock test function.

## F.2    Additional Details on the Experiments and the Source Code

In the paper, we have provided a complete theoretical analysis of sample complexity. Here, we briefly mention the computational complexity of the algorithms. There are two computational bottlenecks in implementing Bayesian optimization algorithms. First bottleneck is the update of the GP model based on past observations which requires an $\mathcal{O}(n^3)$ computation at time $n$, due to the matrix inversion, $(k(X_n, X_n) + \lambda^2 I_n)^{-1}$, step. Sparse approximations of matrix inversion [9] or sparse variational models [10, 11, 12] can reduce the computational complexity from $\mathcal{O}(n^3)$ to $\mathcal{O}(n)$, however at the

price of introducing an approximation error. Second bottleneck is the selection of the observation point based on the *acquisition* functions which are summarized next for each algorithm.

- IGP-UCB: $\mu_n(x) + \beta_n^\delta \sigma_n(x)$ where $\beta_n^\delta = \left( B + R\sqrt{2(\gamma_n + 1 + \log(\frac{1}{\delta}))} \right)$.

- GP-PI: $\Pr[f(x) \geq \mu^+ + \alpha] = \Phi\left( \frac{\mu_n(x) - \mu^+ - \alpha}{\sigma_n(x)} \right)$, where $\mu^+ = \max_{i<n} \mu_{i-1}(x_i)$, $\alpha > 0$ is a user selected hyper-parameter (set to $0.01$ in our experiments as suggested in [13]), and $\Phi$ is the cumulative density function of the standard normal distribution.

- GP-EI: $\kappa\Phi(\frac{\kappa}{\sigma_n(x)}) + \sigma_n(x)\phi(\frac{\kappa}{\sigma_n(x)})$, where $\kappa = \mu_n(x) - \mu^+ - \alpha$, and $\phi$ and $\Phi$ denote the probability density function and cumulative density function of the standard normal distribution, respectively. The parameters $\mu^+$ and $\alpha$ are set similar to GP-PI, following [13].

The standard approach in finding the maximizer of the acquisition function is to evaluate it on a grid discretizing the search space [14]. For a grid of size $M$, this requires $O(Mn)$ computations at time $n$. We have used the same discretization for all algorithms.

A practical idea to improve the computational cost in implementing Bayesian optimization algorithms is to use an off-the-shelf optimizer to solve the optimization of the acquisition function at each iteration (instead of using a grid). This method, although can lead to significant gains in computational complexity, invalidates the existing regret bounds, due to lack of guarantees for an accurate optimization of the acquisition function (that is often non-convex). We thus used the discretization method, following most related work with analytical regret guarantees [e.g., 5, 14]

The source code includes the following files:

- *algorithms.py:* Contains all learning algorithm (IGP-UCB, MVR, GP-PI and GP-EI) classes and the corresponding routines

- *function.py:* Contains the definition of the objective functions including Rosenbrock, Hartman3, and RKHS elements.

- *main_synthetic_test_fct.py, main_rosenbrock.py, main_hartman.py:* each contains the main file which runs the experiment with the corresponding objective function.

In addition, we have provided two files *main_demo_1.py, main_demo_2.py* for demonstration purposes.

At the beginning of each main file a run command example is provided that contains the various keyword arguments which may be passed to the experiment. Those include values such as the number of samples and the learning algorithm. For instance, the run command "*python main_rosenbrock.py -N 100 -n_samples 20 -n_points 200 -methods MVR GP_PI*" runs the experiments on Rosenbrock test function for $N = 100$ exploration trials, for 20 times (the final figures shall be generated by averaging over the output values of these 20 experiments), where a discretization of the domain with $M = 200$ points is used. It will generate a simple regret vector $[r_n^{\mathcal{A}}]_{n=1}^N$, as output values, for each experiment, for all of the specified learning algorithms (MVR and GP-PI in this example), that will be stored as a column in a `.csv` file. See also the *README.md* file provided with the source code.

Our experiments take approximately 100 hours, on four devices with $16, 16, 16$ and $64$ GB RAM, and all with `Intel Core i7` processors.