# OpenReview forum: "Optimal Order Simple Regret for Gaussian Process Bandits"
_NeurIPS.cc/2021/Conference — NeurIPS 2021 Poster_

### Official Review · Reviewer_HA1h · 2021-07-01

**Rating:** 7
**Confidence:** 4

**Summary:**

This paper studies the problem of pure exploration in Gaussian processes. In particular, given a function sampled from a GP, the paper gives bounds on the simple regret of the classifier: the gap between the predicted optimum at time n and the true optimum. The authors assume an additive noise model where the noise is either assumed to be subgaussian or more generally “light tailed” according to a definition provided therein. In these settings, the authors develop novel confidence widths for GPs with additive noise. These are not the first bounds for GPs. Chowdhury and Gopalan for instance give tight confidence widths. What is novel is that in contrast to past works, the authors do not assume that the additive noise is Gaussian- a fact that makes computing past posterior distributions easier. These bounds allow the authors to derive simple regret bounds that scale like O(\sqrt(\gamma_N / N)) matching known lower bounds up to logarithmic factors and shaving a factor of \sqrt(\gamma_N) off of prior art. As the authors note, without this improvement, there are settings where past algorithms are not guaranteed to converge while theirs is. It is worth noting however, that the lower bounds given in 28 and 29 appear to be minimax in nature which would imply that the algorithms presented in this paper are minimax optimal, but that an instance dependent algorithm may be possible. For instance, a very recent work (recent enough that is does not impact my score since that would be unfair) https://arxiv.org/abs/2105.05806 studies pure exploration in RKHSs and gives a instance dependent result. They specialize their result to the GP case, and it may be worth comparing against them?

**Limitations And Societal Impact:**

The authors do a good job discussing the limitations of their work, though more consideration should be given to potential negative societal impacts than simply saying “our work is theoretical, therefore we can do no wrong.”

**Main Review:**

Strengths: The paper is very clearly written and easy to understand which I appreciate as a reviewer. The theoretical results seem strong too. In particular, the confidence widths may be of more practical relevance than the rest of the paper given the growing popularity of GP and RKHS models. The algorithm also improves over prior art in terms of simple regret. I would be curious to know if using these confidence widths with past algorithms could lead to improvements on the order of \sqrt(\gamma_N) as well or if the improvements that this paper shows in its simple regret guarantee are algorithmic as well as due to the confidence bound. The latter is a stronger contribution than the former, especially since MVR sampling exists in previous work.

Weaknesses: There is little improvement empirically. Furthermore, it is unclear if the gains in this paper are due solely to the confidence widths or if the design of the algorithm is important too. For the empirical study, it is unclear how the other experiments would perform if they had access to the same confidence widths presented in this work. This may make the algorithmic comparison fairer since the differences in performance would be solely due to the sampling procedures. Also, (and I am torn on this since the setup is nice and clear) it is worth noting that the authors are most of the way through page 5 before any results are presented.

Other comments and questions:
-	Does theorem 1 hold for an adaptive sequence of x_n’s or a fixed sequence? The theorem just seems to specify a set of (x,y)’s that have been collected. Ie, is this a truly anytime result or for a fixed sequence? In the case of a linear kernel, the gap in the confidence widths between an anytime and fixed confidence bound is O(\sqrt(d)) which behaves like O(sqrt(\gamma_n)) in that setting. I guess that the algorithm is using these as an adaptive sequence which is maybe okay from a Bayesian perspective.
-	Same question for Thm 2
-	For the result in remark 2, do other works get the same factor of d since log(N^d) = dlog(N)? This work is tighter in terms of \sqrt(\gamma) but is the d dependence the same?
-	Why is MVR the right sampling objective?
-	Regarding the statement in Section 6 about simple and cumulative regret bounds, it is somewhat expected that the cumulative regret is linear if you do this well on simple regret as your objective is largely one of exploration. Take for example the SE kernel as the variance \sigma -> 0. In this setting, we recover standard multiarmed bandits where http://sbubeck.com/ALT09_BMS.pdf for instance show that there cannot be an algorithm that is simultaneously optimal in both simple and cumulative regret.

Minor comments:
-	Make sure that the colors chosen for the plots are colorblind friendly. There are a variety of palettes in python for this.
-	Some of the axes in the plots in the main body and especially Appendix G are hard to read.


**Time Spent Reviewing:**

4

---

> ### Author Response · Authors · 2021-08-10
> **Response to Reviewer HA1h**
>
> Thank you for your predominantly positive feedback on our work. We appreciate your thorough review and useful comments. We are also glad that you find the paper clearly written, and the results strong. We also agree that our confidence intervals may be of independent interest for the community. Below we will provide point to point response to your comments.
>
> - *”I would be curious to know if using these confidence widths with past algorithms could lead to improvements on the order of \sqrt(\gamma_N) as well or if the improvements that this paper shows in its simple regret guarantee are algorithmic as well as due to the confidence bound. The latter is a stronger contribution than the former, especially since MVR sampling exists in previous work.”*
>
> The short answer to this question is that the simple regret guarantee is both algorithmic as well as due to confidence intervals.
> As we have mentioned in lines 33-38 of Appendix A, our confidence intervals do not directly apply to the analysis of cumulative regret of algorithms such as GP-UCB. The key difference is that, in MVR, the observation points $x_n$ are independent of the noise terms $\epsilon_n$ (although $x_n$ are allowed to depend on ${x_j}$, ${j<n}$ and $\hat{x}^*_N$ is allowed to depend on ${x_n,\epsilon_n}$, ${n\le N}$),
> while in GP-UCB $x_n$ are allowed to depend on $\{\epsilon_j\}$, ${j< n}$.
>
> It remains an interesting open question whether the state of the art upper bound on the regret performance of GP-UCB [20] is tight or the gap with the lower bound [28,29] is an artifact of its proof.
>
>
> - *“There is little improvement empirically. Furthermore, it is unclear if the gains in this paper are due solely to the confidence widths or if the design of the algorithm is important too. For the empirical study, it is unclear how the other experiments would perform if they had access to the same confidence widths presented in this work. This may make the algorithmic comparison fairer since the differences in performance would be solely due to the sampling procedures.”*:
>
> As mentioned above, confidence intervals developed in Theorems 1 and 2 do not apply to the analysis of GP-UCB (as well as other algorithms whose analysis is inspired by that of GP-UCB [19], such as GP-TS [20] and GP-EI [39] and their variants). Please also see the first 4 items in our response to Reviewer kShU where we contextualize our results in the literature.
>
>
> - *”Also, (and I am torn on this since the setup is nice and clear) it is worth noting that the authors are most of the way through page 5 before any results are presented.”*:
>
> We agree with the reviewer that this one comes down to a choice. We chose to be absolutely clear with our problem formulation and assumptions which take some space. There seems to be a consensus among the reviewers that the paper is clearly written which we partially attribute to this organization.
>
> **Other comments and questions:**
>
> - *”Does theorem 1 hold for an adaptive sequence of x_n’s or a fixed sequence? The theorem just seems to specify a set of (x,y)’s that have been collected. Ie, is this a truly anytime result or for a fixed sequence? In the case of a linear kernel, the gap in the confidence widths between an anytime and fixed confidence bound is O(\sqrt(d)) which behaves like O(sqrt(\gamma_n)) in that setting. I guess that the algorithm is using these as an adaptive sequence which is maybe okay from a Bayesian perspective. Same question for Thm 2”*:
>
> Let us first clarify the definition of an **anytime** bound as we understand it. For a sequence of $\{(x_i,y_i)\}$, $i=1,2,...$, the bound is anytime, if it holds true for $\{(x_i,y_i)\}$, $i=1,2,...,N$, for all $N=1,2,3,...$.
>
> With this definition, our proof of confidence intervals (Theorems 1 and 2) does not show an anytime bound. In contrast it holds for a fixed $N$. Consequently, the regret bound also holds for a fixed $N$. We will clarify this point in the paper.
>
> Please note that the confidence bounds (Theorems 1 and 2) and consequently the regret bound can be extended to anytime bounds using  a simple probability union bound argument. In particular, if we replace $\delta$ in the proof with $\frac{6\delta}{\pi^2n^2}$ which satisfies $\sum_{n=1}^\infty \frac{6\delta}{\pi^2n^2} = \delta$, we obtain anytime bounds with a confidence $1-\delta$. We will add this as a remark.
>
> We hope that we correctly understood and addressed the question. Please let us know if we misunderstood this comment.
>
>
>
> - *”For the result in remark 2, do other works get the same factor of d since log(N^d) = dlog(N)? This work is tighter in terms of \sqrt(\gamma) but is the d dependence the same?”*:
>
> The short answer is no. Our results are worse than, e.g., those of Chowdhury and Gopalan [20] by a factor of $\sqrt{d}$ (up to logarithmic factors in $N$). Thank you for poitning this out. We will make it clear in our presentation.
>
> Roughly speaking, here we are trading off a square root of $\gamma_N$ for a square root of the input dimension $d$ (up to logarithmic factors in $N$). This is the gain which allows us to show sublinear regret bounds in $N$; since $\gamma_N$ may grow polynomially with $N$ while $d$ is fixed.
>
> - *"Why is MVR the right sampling objective?"*
>
> The main motivation here is analytical. We choose an algorithm that is amenable to a tight regret analysis based on our Proposition 1 and Theorems 1 and 2. Please also see our response to reviewer 1 regarding contextualizing our work in the literature.
>
> - *”Regarding the statement in Section 6 …”*:
>
> Thanks for pointing out Bubeck *et al*’s result on the contrast beteen the simple and cumulative regrets. Their result provides interesting insights. We will reflect your comment on this point in the paper.
>
>
> *Minor comments*:
>
>
> Thanks for the suggestions. We will make sure that plots are colorblind friendly. We will also increase the size of the figures in the camera ready version where an additional page is allowed
>
>
> **The related work which is posted to ArXiv recently**
>
>
>
> Thank you for pointing out the recent work (https://arxiv.org/abs/2105.05806) which was published on ArXiv a few days before the abstract submission deadline. We were not aware of this work. They are tackling a similar problem to ours. This is encouraging in showing that our formulation and results are interesting for other researchers in the community.
>
> Part of their contribution seems to be on a particular formulation of misspecified models where the true $f$ is not in the RKHS but the $L^\infty$ norm of $f-g$ is bounded by a constant $h$ and $g$ is in the RKHS. Regardless, when $h=0$ their model recovers the standard RKHS setting.
>
> - Their regression method is different from the standard kernel ridge regression and GP models.
> - Their algorithm is based on arm-elimination techniques in the $K$-armed bandit literature which dates back to [Auer and Ortner 2011, "UCB Revisited ..."). They assume a finite number of arms ($|\mathcal{X}|<\infty$). Their algorithm proceeds by eliminating suboptimal arms in successive rounds. A discretization argument however can perhaps be used to extend their results to a continuous domain.
> - If we understand it correctly, the **instance dependent** bounds refer to the bounds which are based on the gap between the best and second best inputs to $f$ (or the best and second best arms in the $K$-armed bandit problem), which is a well-posed problem only on finite sets and not on a continuous domain setting (where the gap is always $0$, assuming a continuous function).
> - Their Theorem 3 seems to provide a state of the art regret bound. We will cite this result.
> - While the setup of their Theorem 4 on the sample complexity is similar to our Corollary 1, we are unable to make a conclusive comparison for the reasons explained below.
>
> 1. First of all their results are not explicitly given in terms of $\epsilon$. In fact, their results are subject to solving a nonlinear equation involving an implcitely given function $g(\epsilon)$.
> 2.In addition, their Theorem 4 does not hold for all values of $\epsilon$, but only for the values of $\epsilon$ larger than a certain $\bar{\epsilon}$. As it is later remarked on the second column of page 8, to make $\bar{\epsilon}$ small, their regularization parameter $\gamma$ should be small. However, since the relation between these two is not characterized, and since the solution to the nonlinear equation involving $g$ is not explicitly given, it is not possible for us to directly compare the sample complexities. They however mention that for the special case of linear kernels $k(x,x') = x^{T}x'$, ${H}_k= \mathbb{R}^d$, they recover optimal order sample complexities (up to log factors).
>
> We will include this discussion in the camera ready version of our paper if it is accepted.
>
> We appreicate the amount of time that is gone into this review.

---

> > ### Comment · Reviewer_HA1h · 2021-08-31
> > **Thank you**
> >
> > Thank you for responding to my questions and clarifying some of your results for me, especially regarding anytime bounds. My main confusion was regarding fully adaptive sequences, and I should have said this more clearly. For instance, if one is computing the least squares estimator in linear bandits, different confidence widths are needed for an algorithm like lin-UCB (which has a full adaptive sequence of arm pulls) versus an algorithm like RAGE (which uses a fixed design in rounds). The latter avoids a factor of sqrt(d) in the confidence width as a result. I was wondering about this in your setting, and you've largely addressed it. In any case, I am happy to increase my score.

---

> > > ### Author Response · Authors · 2021-09-02
> > > **Thank You**
> > >
> > > Thanks. We appreciate your comprehensive review and positive feedback on our work. We will use several points from this review to improve the paper (including the remark on anytime bounds, the remark related to Bubeck et al’s result, and the comparison with the arXiv paper: Camilleri et al'21).

---

### Official Review · Reviewer_wjVJ · 2021-07-16

**Rating:** 7
**Confidence:** 4

**Summary:**

This paper considers the Gaussian process optimization problem where the objective function f lives in a reproducing kernel Hilbert space (RKHS). So far there is a significant gap between the lower and upper bound on the simple regret performance. The authors propose a full exploration algorithm, called Maximum Variance Reduction (MVR) which achieves a simple regret bound $\mathcal O^*(\frac{\sqrt{\gamma_T}}{T})$, where {\gamma_T} is the maximum information gain. This bound significantly improves the existing simple regret bounds. Further, on specific Square Exponential and Mat\’ern kernels, their bound matches the lower bound up to logarithmic factors. Further, the authors extend their results to the more general class of light-tail distributions, thus broadening the applicability of the results. Finally, they validate their theoretical results on several synthetic functions and show that their algorithm is better than other algorithms such as GP-PI, GP-EI, IGP-UCB.

**Ethical Concerns:**

I do not see any ethical concerns.

**Limitations And Societal Impact:**

Yes.

**Main Review:**

The paper is well written and easy to follow. A central result of their theoretical analysis is to provide the novel and sharp confidence intervals for GP model in the RKHS space. This key result is obtained via a novel connection between Gaussian processes regression and Kernel ridge regression.  Under the theoretical view, this is a good paper. The results are significant especially these novel confidence intervals. While simulations are a bit limited given that we are looking at a new algorithm, I do not see it as much limitation as the key focus here is theory.

Question: In all experimental results, why the graphs of MVR always decrease while the graphs of other baselines are not stable?

Some minor mistakes:
1.	The formula of EI acquisition at line 212 seems incorrect (the argument of function \Phi should be \sigma/\kappa instead of \kappa/\sigma). If so, please fix it!
2.	At the second equation in the proof of Lemma 1, t should be n.


**Time Spent Reviewing:**

6

---

> ### Author Response · Authors · 2021-08-10
> **Response to Reviewer wjVJ**
>
> Thank you for your positive feedback on our work. We are glad that you find the paper well written, and the results significant. We hope you could kindly share your interest in the paper with the other reviewers in the upcoming discussion.
>
>
> *Question: In all experimental results, why the graphs of MVR always decrease while the graphs of other baselines are not stable?*
>
> This seems to be due to the fact that the observation points $x_n$ in GP-UCB, GP-PI and GP-EI randomly change based on the noise in the past observations. While in MVR $x_n$ are independent from the noise in the past observations. Ideally, for a very large Monte Carlo sample all the graphs should concentrate around their mean. That unfortunately requires a computation budget beyond ours. For the information on the details of the devices and computation time please see Appendix G.2.
>
> Thank you very much for bringing the typos in lines 66 and 212 of the supplementary material to our attention. We will fix both typos and have another careful proofread of the supplementary material. We appreciate the amount of time that has gone into the review.

---

### Official Review · Reviewer_yRtV · 2021-07-17

**Rating:** 7
**Confidence:** 3

**Summary:**

The paper derives improved upper bounds for the best-arm identification problem in GP bandits. Specifically, they improve the previously best-known upper bound on simple regret from O(\gamma_N/\sqrt{N}) to O(\sqrt{\gamma_N/N}), where \gamma_N is the maximal information gain and N is the number of function evaluations.

Removing the extra \sqrt{\gamma_N} factor is important as \gamma_N can grow faster than \sqrt{N}. In particular, for the special cases of Matern and SE kernels, the authors show that the new bound (unlike old ones) goes to 0 as N-->\infty and is only logarithmically far from previously established lower bounds.

The improvement in regret analysis boils down to a new confidence ellipsoid bound that is available for both subgaussian and light-tailed noise distributions. Unlike previous works, that focused on cumulative regret, the new confidence ellipsoid bound requires that query points x_1,...,x_n are independent of noise terms e_1,...,e_n for each n. This additional restriction gives improved bounds and is not restrictive for best-arm identification using a maximum variance reduction algorithm.

**Limitations And Societal Impact:**

This is a theoretical work. Limitations of the analysis is clearly presented.

**Main Review:**

To the best of my knowledge, the author's result is new and shows clear improvement to state-of-the-art for important problem of best-arm identification in GP bandits. Comparison to related literature is transparent and nicely presented. Also, the paper is very well-written and I believe is accessible and will be of interest to broad audience. Thus, I recommend acceptance.

One remark:
You mention a few times that "simple regret is favorable in situations with a preliminary exploration phase". It might be worth elaborating a bit on this point. A concrete example would benefit the reader distinguish between the relevance of simple vs cumulative regret.

**Time Spent Reviewing:**

2

---

> ### Author Response · Authors · 2021-08-10
> **Response to Reviewer yRtV**
>
> Thank you for your positive feedback on our work. We are glad that you find the paper well written, and accessible and of interest to a broad audience. We also believe that the results may be interesting for the researchers in bandit, GP and learning communities. We hope you could kindly share your interest in the paper with the other reviewers in the upcoming discussion.
>
> - *One remark: You mention a few times that "simple regret is favorable in situations with a preliminary exploration phase". It might be worth elaborating a bit on this point. A concrete example would benefit the reader distinguish between the relevance of simple vs cumulative regret.*:
>
> Thanks for your suggestion. In the paper, we use the example of hyperparameter tuning in machine learning and artificial intelligence models. Here, all possible configurations of hyperparameters correspond to $\mathcal{X}$ and a measurable performance metric such as the error rate is considered as $f$. During the preliminary exploration phase, $N$ configurations of hyperparameters are tried out. When the final configuration is decided, those hyperparameters will be used in the model. In contrast, in the cumulative reward settings, the performance of intermediate steps are also taken into account in the evaluation. In this example, simple regret seems a better choice for the performance measure. Following your comment, we will add more details on this exampe to the paper.
>
> As another example, we can consider an industrial design problem. The posssibilities for the parameters of the design correspond to $\mathcal{X}$, and a measurable performance metric of the design is considered as $f$. The goal is to try $N$ configurations of the design parameters as an initial exploration phase. When the final design is decided, it will be used for production. This is another example where there is a clear distinction between the initial exploration phase of designing which is limited by available resources and the deployment phase that comes after that. In contrast, in a cumulative regret setting, the design is updated during the deployment. An example for this latter case is online A/B testing used for example in online shopping.

---

> > ### Comment · Reviewer_yRtV · 2021-09-01
> > **Thank you for your response**
> >
> > Thank you for your response. I would suggest discussing these examples in the camera ready using the additional one page provided to you. I keep my positive score for the paper.

---

> > > ### Author Response · Authors · 2021-09-02
> > > **Thank You**
> > >
> > > Thanks. We will add the example of industrial design and extend on our discussion of the example of hyperparameter tuning. We appreciate your positive feedback on our work.

---

### Official Review · Reviewer_kShU · 2021-07-18

**Rating:** 5
**Confidence:** 5

**Summary:**

The paper introduces a new approach for simple regret minimization in Bayesian optimization. This is a useful testbed as an intermediate setting harder than pure black-box optimization due to the presence of noise, but avoiding some of the worst case pitfalls of a cumulative regret analysis.

Note that a GP with noisy observation will use regularization in its posterior, and that standard analysis is to split the prediction error in a part depending on the regularized posterior as if it was trained on the noiseless feedback and an error term as if it was trained only on the noise. From this template the paper derives its two main novel approaches:
- Prop. 1, a new equality that connects the maximum (worst-case) prediction error of a regularized posterior trained on (unavailable) noiseless observation to the posterior variance of the regularized GP plus a regularization term depending on the regularization (and therefore on the noise level)
- Thm. 1, where the main difference is that using Prop. 1 the authors construct a point-by-point concentration inequality rather than the more commonly martingale concentrations used in the literature that apply to the whole input space. The upside is that this allows for confidence intervals whose radius does not depend on the information gain $\gamma_N$, unlike most methods. The downside is that it requires the noise to be uncorrelated with the observations, which in optimization means that the candidates for evaluation are chosen without looking at the feedback at all. The authors also provide a variant (Thm. 2) for light-tailed noise.

With these two tools the regret analysis is pretty straightforward, relying on an ideal discretization of the input space to extend the for-any concentration to a for-all bound, and then union bound the success probability. The final tool required is a unsupervised strategy to select candidates, and the authors resort to greedy information gain maximization, playing a sort of optimism role in the inner optimization loop. Finally all the candidates are evaluated and the best candidate found is returned.

Compared to other simple regret minimization approaches, this shaves a $\sqrt{\gamma_N}$ factor off, taking it closer to its min-max lower bound.

Experimentally, the proposed approach does not seem to outperform existing methods, slightly weakening the argument that the tighter confidence interval improves exploration/exploitation trade-offs in practice.

**Limitations And Societal Impact:**

As a theory paper the direct societal impact is not a direct concern.

**Main Review:**

Overall I feel that the paper is an interesting contribution, but is held back by placing several crucial comparisons in the appendix, painting a picture in the main paper that is not fully satisfactory. For this reason I would like the paper to more clearly state what is a true improvement and what is not.

The phrase L66:"It is noteworthy that our bound guarantees convergence to the optimum value off, while previous ̃$O(\gamma_N/\sqrt{N})$ bounds do not" is quite misleading. There are at least two other algorithms that match MVR's regret rate. SupKernelUCB already achieves the improved $O(\sqrt{\gamma_N/N})$ rate, and is only mentioned in the appendix, while it should be the most obvious baseline. pi-GP-UCB is also guaranteed to achieve at least sub-linear regret, but is only cited and not discussed in the main paper.
Elements brought in favour of MVR in the appendix are 1) generality, 2) simpler implementation  and 3) working efficiently in practice (L28)
1) Regarding generality, SupKernelUCB holds for any kernel just like MVR. pi-GP-UCB is specific to the Matern kernel, but note that also MVR's analysis require a good discretization that is not easy to find for kernel that are not translation invariant. Regarding the restriction
of SupKernelUCB's analysis only to finite sets, both Janz et al. [21] and Cai et al [29, Appendix A.4] argue that it can be easily extended to continuous settings with a discretization argument similar to the one used for MVR. Moreover in practice running MVR with guarantees might also requires a discrete set (more of this later) and not being so easy to apply to continuous set.
2) Regarding simplicity of implementation, even a single iteration of the MVR optimization problem (which is known in the literature as greedy information gain maximization) is a well known NP-HARD problem on continuous sets, since it requires optimizing a non-convex function. In practice, this essentially voids the regret guarantees for MVR, unless the domain is finite where we can find the optimum by enumeration. Comparing the complexity of MVR and pi-GP-UCB, MVR is indeed simpler but pi-GP-UCB is not much harder to implement than IGP-UCB (it is essentially IGP-UCB + a tree partitioning of the space), and IGP-UCB is included in the experimental comparison.
3) Regarding working efficiently in practice, the plots are quite inconclusive. If all the algorithms converge quickly before 100 iterations, it makes little sense to plot up to 400 iterations with markers every 50. It would rather be more useful to zoom in the first 100 iterations, and increase the number of repetitions to observe convergence speed.

Overall I find the paper quite interesting, and a nice bridge between online simple regret optimization and active learning. However the lack of comparison with its most obvious baselines, no statistically significant improvement even on toy data, and several minor exposition problems (more in detail later) push it slightly under the bar. I am open to reconsider my score if a more rigorous comparison is included where appropriate in the main paper rather than the appendix, showing clearly where MVR outperforms existing state of the art methods.

MINOR:
-The probability delta for a single sample and delta for the overall algorithm should be probably indicated with different symbols. It took me a long time to understand why large discretizations would negatively impact the regret, until i noticed that the denominator of the beta function becomes the numerator of the logarithm.
-It might be good to include a remark highlighting that the algorithm needs to know both a bound on the function norm and on the noise level, which are usually hard to estimate. This is ok and common in the frequentist literature, but under bayesian assumptions the norm of the function can be unknown.
- It would be good to include in the appendix one or two examples of how to generate covering grids that satisfy the assumptions (and if the method used in the experiments do so).

**Time Spent Reviewing:**

8

---

> ### Author Response · Authors · 2021-08-10
> **Response to Reviewer kShU**
>
>
> Thank you for your thorough review and useful comments. We appreciate the amount of time that has gone into this review.
>
> We are glad that you find our contribution interesting. We also regret that minor issues around the discussion of related work prevented you from having as pleasant a reading experience as our other reviewers. The main criticism seems to be on the comparison with the existing work, in particular, SupKernelUCB and $\pi$-GP-UCB. We first reiterate the positioning of our work in the landscape of the literature. We then address the comments point by point especially with respect to the comparison with SupKernelUCB and $\pi$-GP-UCB.
>
> - Our focus here is on the theoretical analysis of the GP bandit problem, and finding its performance limits, in the RKHS setting, and in the presence of observation noise. We would like to point out that there are numerous classic GP bandit algorithms such as GP-UCB, GP-TS, GP-EI, GP-PI, and their variants. The regret analysis for GP-TS [20] and GP-EI [49] are inspired by the pioneering analysis of GP-UCB [19] (we are not aware of any analysis for GP-PI especially in our setting). The regret bounds for these algorithms (as well as their variants in other settings [34-45]; see lines 95-98 of the paper) show an $\tilde{O}(\gamma_N/\sqrt{N})$ regret bound. This bound unfortunately is not always diminishing as $N$ grows; since $\gamma_N$ can grow as fast or faster than $\sqrt{N}$. Thus, the existing regret bounds for these algorithms are trivial in many cases of interest. Examples include Matérn kernels with certain smoothness parameters, e.g., Laplace kernel and neural tangent kernel (which has the same behavior as Laplace kernel as shown recently) as special cases of the Matérn family of kernels.
> - We emphasize that it is unclear whether this suboptimal regret bound is a fundamental shortcoming of GP-UCB (also GP-TS, GP-EI, and their variants as mentioned above) or an artifact of its state of the art analysis (based on self-normalized martingale inequalities of [20]). This remains an interesting open problem to be addressed (see lines 33-38 in Appendix A).
> - Thus, from a theoretical perspective, this seems a very interesting and important problem for the learning and bandit communities that whether “always sublinear” and “order optimal” regret bounds are achievable for this problem, yielding finite sample complexity to reach certain optimization precision (either in expectation or with high probability). In particular, it is interesting to see whether $\tilde{O}(\sqrt{\gamma_N/N})$ diminishing regret, which matches the lower bound, for the cases where the lower bound is known [28,29], is achievable or not.
>
> - Unable to address this open problem for GP-UCB (also GP-TS, GP-EI and their variants), and similar to several recent works in the literature (see, e.g., $\pi$-GP-UCB and LP-GP-UCB; references [8] and [9] in Appendix A), we have investigated the feasibility of optimal order learning for other algorithms rather than commonly used GP-UCB, GP-TS, GP-EI and GP-PI. In particular we consider MVR which is amenable to a tight regret analysis based on our Proposition 1, and Theorems 1 and 2. As correctly mentioned by the reviewer, two algorithms deserving significant attention are perhaps SupKernelUCB and $\pi$-GP-UCB.
>
> **Comparison with SupKernelUCB and $\pi$-GP-UCB:**
>
> - Compared to $\pi$-GP-UCB [21], our analysis of MVR shows a significantly tighter regret bound. We make this comparison on the Matérn family of kernels because $\pi$-GP-UCB is specially designed for this family of kernels. In [21, Theorem 3] an upper bound on the regret of $\pi$-GP-UCB is provided which in our notation implies $r^{\pi-{GP-UCB}}_N =\tilde{O}(N^{\frac{-2\nu-d}{4\nu+d(2d+4)}})$. In contrast, we prove $r_N^{MVR} = \tilde{O}(N^{\frac{-\nu}{2\nu+d}})$ that is uniformly (in all values of $\nu\ge 0.5$ and $d\ge 1$) tighter than $r_N^{\pi-GP-UCB}$. In addition, MVR has a simple pure exploration structure while $\pi$-GP-UCB is quite involved by partitioning the input domain to increasingly many hypercubes and fitting an independent GP model to each hypercube.
> - Compared to SupKernelUCB (reference [6] in Appendix A), our regret bound is tighter with a factor of $\log^{\frac{3}{2}}(N)$, when we take into account the effect of discretization argument for SupKernelUCB. We could not find a complete proof for the performance of SupKernelUCB in continuous domains as a reference. We rely on the remarks of Janz et al. [21] and Cai et al. [29, Appendix A.4] for making this comparison.
> - The same papers [21,29] mention that SupKernelUCB is an impractical algorithm which is only of theoretical interest. This is due to its complex structure which partitions the past observations into independent batches (a technique similar to the one used in the analysis of SupLinUCB [Chu et al., 2011]). This batching technique yields loss of information which seems the key contributing factor to the poor performance of SupKernelUCB in experiments and that is independent of the discretization argument.
> - In fact, as much as it comes to the need for discretization, all algorithms (GP-UCB, GP-TS, GP-EI, MVR, SupKernelUCB, $\pi$-GP-UCB) share the same issue: the optimization of the acquisition function which is a non-convex, and possibly NP-Hard optimization problem as correctly mentioned by the reviewer. That leads to a high **computational complexity** but does not invalidate the regret bounds (or bounds on the **sample complexity**) from a learning theory perspective (please find more on this below).
>
> **1. "Regarding generality ..." :**
> As mentioned above, the key contributing factor to the poor performance of SupKernelUCB seems to be the loss of information due to the independent batch sampling structure. Regarding the discretization argument, please notice that the use of discretization in *the analysis of the regret bound* is completely separate from the need for discretization *for the optimization of the acquisition function*, as discussed below.
>
> i) As an analytical technique, we know such discretization exists and we use it to extend the pointwise confidence intervals to uniform (in input x) ones. This is not related to implementation.
>
> ii) On the other hand, all of the acquisition based algorithms (GP-UCB, GP-TS, GP-PI, GP-EI, MVR, SupKernelUCB, $\pi$-GP-UCB) need to optimize an acquisition function. The standard analytical approach is to neglect the cost of this optimization (the same for all the existing work to the best of our knowledge) and assume that the acquisition function can be efficiently optimized. The rationale behind this approach to GP bandits is that the optimization of $f$ is based on expensive observations from environment (the performance of a design, the error rate of a machine learning model for certain hyper parameter configuration), while optimization of the acquisition function is a purely computational matter. This is a direct quote from the celebrated work of Srinivas et al [19] on GP-UCB: “If D [domain] is infinite, finding $x_t$ [$x_n$ in our notation] in (6) may be hard: the upper confidence index is multimodal in general. However, global search heuristics are very effective in practice (Brochu et al., 2009). It is generally assumed that evaluating $f$ is more costly than maximizing the UCB index.”
>
> *Remark*: in practice we can use an off the shelf optimization library to optimize the acquisition function. This however invalidates the regret bounds due to the lack of guarantees for the non-convex optimization problem. In analytical literature thus it is standard to assume a perfect optimization of the acquisition function or rely on a discretization of the domain to optimize the acquisition function. Please notice that even with a perfect optimization of the acquisition function, the GP bandit problem is still difficult. As we mentioned above, there are still open problems with regards to the analysis of even simple algorithms such as GP-UCB.
>
> In summary, the need for discretization to optimize the acquistion function or assuming a perfect optimization of the acquistion function is not specific to our results and is the standard approach in the literature. We will add this remark in the paper.
>
> **2. "Regarding simplicity of implementation ...":**
> Please see the answer above. Also, as it was mentioned, $\pi$-GP-UCB partitions the domain into increasingly many hypercubes and fits an independent GP model to each hypercube which makes its implementation tedious in comparison to other algorithms.
>
> **3. "Regarding working efficiently in practice ...":**
> Thanks for your suggestion. We will amend the graphs according to your comment. With more space available in the camera ready version, if the paper is accepted, we will increase the size of the graphs for better visibility.
>
> **Other comments:**
>
> - *"I am open to reconsider my score if a more rigorous comparison is included where appropriate in the main paper rather than the appendix, showing clearly where MVR outperforms existing state of the art methods."*:
>
> For the clarity of exposition, it seems appropriate to have the technical comparison with some exisitng work including $\pi$-GP-UCB, LP-GP-UCB and SupKernelUCB after a rigoruous problem formulation and the introduction of notations. If the paper is accepted, we will have an extra page for the camera ready version, where **we will move this discussion from Appendix A to subsection 4.4 of the main paper**. We will include a clear comparison adding more details as given above.
>
> - *"MINOR"*:
>
> Thanks for your suggestions. We will amend the paper accordingly. In particular, we will include a remark on the knowledge of the RKHS norm of $f$ and the sub-Gaussianity parameter of noise, which are standard in the literature. We will also comment on the covering grid.

---

### Decision · Program_Chairs · 2021-09-27

**Decision:**

Accept (Poster)

**Comment:**

Based on the reviews and discussion, the consensus is that the contributions of the paper are valuable and worth publication in NeurIPS, particularly the new confidence bounds leading to a simple proof of order-optimal simple regret.  The initial concerns were largely resolved following the author response.   One reviewer score remains below the threshold, but unfortunately that reviewer did not respond to the rebuttal nor take part in any of the discussions, so I will not put as much weight on their score, even with the highest-confidence rating.  The reviewing team did, however, discuss that reviewer's concerns, and came to the consensus that they serve as points that should be addressed in the final version, but not serve as a reason for rejection.  Specifically:
- Please carefully move the comparisons to the most related earlier works (e.g., pi-GP-UCB, SupKernelUCB) to the main body and expand them.
- Ideally provide at least a brief mention of the issue of being able to optimize the acquisition function exactly.
- Comparing experimentally to pi-GP-UCB would have been ideal, but was not judged to be essential.  Nevertheless, the experiment section could be revised, adding more data points in the plots for t in [0,50], and discussing the results more (e.g., yours vs. IGP-UCB).

Beyond these points, please incorporate the other reviewer comments carefully, and additionally be careful of typos, e.g., (i) the arguments to k(.,.) in Appendix C and (ii) several missing E[.] operations after Line 87 of the supplementary material. These are both central to the analysis, so additional careful proof-reading is strongly encouraged.